psychology

psychiatry, anxiety disorders, mood disorders, cognitive distortions, self-report questionnaire, catastrophizing

**Author for correspondence:**
Alexandra C. Pike
e-mail: alex.pike@ucl.ac.uk

†Joint first authors.

# The development and psychometric properties of a self-report Catastrophizing Questionnaire

Alexandra C. Pike†, Jade R. Serfaty† and Oliver J. Robinson

Anxiety Lab, Neuroscience and Mental Health Group, Institute of Cognitive Neuroscience, Alexandra House, 17–19 Queen Square, London WC1N 3AR, UK

 ACP, 0000-0003-1972-5530; JRS, 0000-0001-6939-2952; OJR, 0000-0002-3100-1132

Catastrophizing is a cognitive process that can be defined as predicting the worst possible outcome. It has been shown to be related to psychiatric diagnoses such as depression and anxiety, yet there are no self-report questionnaires specifically measuring it outside the context of pain research. Here, we therefore develop a novel, comprehensive self-report measure of general catastrophizing. We performed five online studies (total $n = 734$), in which we created and refined a Catastrophizing Questionnaire, and used a factor analytic approach to understand its underlying structure. We also assessed convergent and discriminant validity, and analysed test–retest reliability. Furthermore, we tested the ability of Catastrophizing Questionnaire scores to predict relevant clinical variables over and above other questionnaires. Finally, we also developed a four-item short version of this questionnaire. We found that our questionnaire is best fit by a single underlying factor, and shows convergent and discriminant validity. Exploratory factor analyses indicated that catastrophizing is independent from other related constructs, including anxiety and worry. Moreover, we demonstrate incremental validity for this questionnaire in predicting diagnostic and medication status. Finally, we demonstrate that our Catastrophizing Questionnaire has good test–retest reliability (intraclass correlation coefficient = 0.77, $p < 0.001$). Critically, we can now, for the first time, obtain detailed self-report data on catastrophizing.

## 1. Introduction

Catastrophizing, colloquially defined as imagining or predicting the worst possible outcome, was a term originally used by Ellis

to describe instances where individuals see any negative outcome as 'catastrophic' and intolerable, when in fact it is likely to be worse in anticipation than in reality [1]. A catastrophe, in turn, is defined as 'a momentous tragic event ranging from extreme misfortune to utter overthrow or ruin' [2]. Ellis' description of catastrophizing is closely related to Beck's cognitive distortion of *magnification*, defined as 'inflation of the magnitude of [one's] problems and tasks'. Beck refined this definition in 1979, as when an individual 'always think[s] of the worst. It's most likely to happen to [them]' [3]. These early definitions highlight the fact that catastrophizing has two facets: the prediction of an objective catastrophe, which a majority of others would agree to be disastrous (e.g. the death of a partner or oneself), or the (subjective) perception of a particular negative outcome as 'catastrophic', such as social rejection. One of these is a belief in the excessive likelihood of catastrophic events, and the other is attributing excessive gravity to a situation. Both of these facets often occur together, as is perhaps evident from the fact that they are referred to under the same name—an individual overestimates the likelihood of a negative event, and also believes that the negative event will be catastrophic. Catastrophizing, for the purposes of this paper, is considered to be a cognitive process, which is more common in clinical populations (e.g. those suffering from anxiety or depression), but also exists in the general population. Catastrophizing is also likely to cause distress to the individual experiencing it, and—if this is in the context of a psychiatric diagnosis—may be considered a symptom. We do not consider it to be a diagnosis, but rather a transdiagnostic factor which acts as a predisposing or maintaining factor for anxiety, depression and perhaps other disorders.

Notably, Gellatly & Beck [4] have also suggested that the cognitive process of catastrophizing is transdiagnostic, and present in disorders ranging from phobias, to obsessive–compulsive disorder, to post-traumatic stress disorder, to traumatic brain injury. The role of catastrophizing in psychiatric disorders may in fact be causal: catastrophic misinterpretations of (often somatic) signals have been considered to be a key factor in the onset of panic disorder [5]. Panic disorder patients are more likely to misinterpret sensations such as heart palpitations as the onset of a heart attack or 'insanity', which subsequently leads to escalating panic and panic attacks [6].

Catastrophizing and other cognitive distortions are targeted for treatment in cognitive behavioural therapy (CBT). CBT is based on the theory that attitudes or assumptions developed from previous experience may become distorted, leading to dysfunctional cognitions and negatively biased information processing, resulting in a feedback loop [3,7]. The therapeutic techniques of CBT are designed to identify, test, and correct both the cognitions and the underlying beliefs, leading to symptom reduction [3]. Catastrophizing is targeted in CBT using an approach known as decatastrophizing [8], which is used in the treatment (and prevention) of many varied disorders. Despite the importance of catastrophizing, our ability to measure it in the context of mental health is surprisingly limited.

Catastrophizing has been most thoroughly studied in the field of pain research, which may be partly due to the existence of self-report measures specific to pain catastrophizing. Specific measures such as the Pain Catastrophizing Scale (PCS) [9] and the Coping Strategies Questionnaire (CSQ-CAT) [10] have been developed. Research using these scales has shown that catastrophizing is an important psychological predictor of pain experience [9], such that those who score highly on a measure of pain catastrophizing report more intense pain and often fail to improve with treatment [8,9,11,12].

Importantly, pain catastrophizing has also been shown to be related to psychiatric diagnoses such as depression and anxiety. Higher levels of pain catastrophizing have been shown to be present in both worriers [13,14] and those with elevated levels of anxiety [15,16], and depression [14,16]. CBT treatment studies indicate that changes in pain catastrophizing (measured using the PCS) and negative affect have a positive relationship with changes in clinical pain (such that decreases in catastrophizing decrease reported pain, and vice versa) and the effects of the treatment last for months or years [12,17].

Research into general catastrophizing has generally relied either on the scales developed to measure catastrophizing in pain, structured interviews, or self-report measures that do not focus on catastrophizing or contain few items measuring it. The structured interview procedure most commonly used [13,15] is the 'Catastrophizing Interview' developed by Vasey & Borkovec [18] or a variant of it, in which the participant generates a sequence of catastrophizing steps and rates their likelihood (essentially the opposite of the decatastrophizing method typically employed in CBT). In theory, and as shown in Vasey & Borkovec's study [18], worriers generate more steps than non-worriers and the contents of those steps are more catastrophic. This interview is well validated, and the only existing measure specific to catastrophizing outside of the context of pain research. However, there are a few limitations to this approach. The experimenter who administers the catastrophizing interview is not blind to diagnosis, there are no objective criteria for rating the 'catastrophic' nature of

steps generated by individuals, and requiring participants to generate steps may induce a demand characteristic (such that they generate more steps than they would otherwise). Finally, this interview requires in-person testing, which is not feasible when studying catastrophizing in larger sample sizes, or remotely. Other studies [14,16] use self-report questionnaires which cover a variety of different cognitive distortions, with limited numbers of items focusing specifically on catastrophizing, reducing their sensitivity. In particular, shorter questionnaires do not allow a sufficient range of scores for improvement or deterioration to be tracked over time (as might be required if one wished to assess the effects of decatastrophizing), and also prohibit more nuanced analyses of the structure of catastrophizing (i.e. the underlying factor structure). More specifically, existing self-report questionnaires which contain catastrophizing items include the Cognitive Emotion Regulation Questionnaire, or CERQ [19], which includes four items to assess catastrophizing, all of which are focused on the magnification of past events, in contrast with our conceptualization of catastrophizing as being future-oriented. Another questionnaire, the Cognitive Distortions Scale [20], contains two items to measure catastrophizing (one in a social, and one in an achievement setting), but relies on participants' abilities to generalize from a vignette to other situations in everyday life. Other questionnaires containing items that measure catastrophizing include the Children's Negative Cognitive Error Questionnaire [21] and the Children's Cognitive Style Questionnaire [22], which are not validated for use in adults. Finally, research into catastrophizing in psychiatry may simply use pain catastrophizing scales, which are not validated for use outside of the context of pain research; and are not necessarily designed to (or able to) differentiate between other cognitive distortions or psychiatric phenomena and catastrophizing. For instance, Keefe et al. [11] found that in patients with rheumatoid arthritis, there was a level of construct redundancy in the CSQ-CAT between catastrophizing and depression, with one predicting the other. Furthermore, negative mood and pain catastrophizing were found not to be independent in another study [23]. Sullivan et al. [9] examined scales measuring pain-related catastrophizing, depression and anxiety and found that they were markedly similar in content, although intended to measure distinct constructs. Notably, however, the authors concluded that catastrophizing is an independent construct, but that operational or measurement confounds may have resulted in this apparent redundancy [9].

As noted above, no specific self-report measure of catastrophizing currently exists outside of pain research. Developing a questionnaire that (i) measures catastrophizing outside of the specific context of pain, and is (ii) designed to differentiate between catastrophizing and other constructs such as anhedonia or worry would allow us to increase our understanding of catastrophizing. Such a questionnaire would help researchers and clinicians to better understand the role of catastrophizing in mediating emotional reactions and treatment outcomes [20], and would allow investigation into the role of catastrophizing as a risk factor for anxiety and depression. A questionnaire measuring catastrophizing would also allow CBT and decatastrophizing research to measure changes in catastrophizing more directly, and with greater sensitivity. Personalized medicine approaches could also be developed, where those who score highly on measures of catastrophizing could receive more intensive decatastrophizing interventions. Finally, a catastrophizing questionnaire would allow the investigation of differences in patterns in catastrophizing across individuals (frequency and intensity). The aim of the present study, therefore, is to create a new self-report questionnaire measuring catastrophizing, which can be used in a psychiatric or mental health setting and in the general community.

# 2. Methods

## 2.1. Procedure

Five studies were performed, using iterative versions of the Catastrophizing Questionnaire. In all studies, participants were recruited online, and completed the latest version of the Catastrophizing Questionnaire. We also collected additional demographic information and information about psychiatric diagnosis and medication.

In Studies 1 and 2, we refined an initial 31-item version of the Catastrophizing Questionnaire into a 24-item final version, while also performing exploratory factor analyses to understand the structure of the questionnaire. In Study 3, we assessed the factor structure of this novel instrument using a confirmatory factor analysis (CFA), and also investigated convergent and discriminant validity. To start assessing clinical relevance, we also tested whether scores on the Catastrophizing Questionnaire were able to predict psychiatric diagnosis or psychiatric medication use over-and-above the PHQ-9 and GAD-7,

two commonly used clinical instruments. We also introduced a short version of the Catastrophizing Questionnaire. In Study 4, we examined its test–retest reliability. Finally, we report further evidence for discriminant validity in Study 5.

## 2.2. Participants

The questionnaire versions were hosted online using Gorilla (https://gorilla.sc) and participants were recruited via Prolific (https://prolific.co). Participants were required to be fluent in English, and aged 18–60, without any self-reported history of mild cognitive impairment or dementia. We asked participants to report whether they had any psychiatric diagnosis, and if they were taking any psychiatric medication. These questions were used as additional metrics of convergent validity, not as exclusion criteria. Demographic information on participants from all studies is reported in the electronic supplementary material.

We did not perform a power analysis to determine the sample size for these studies, as the issue of how best to perform a power analysis for factor analysis research is still an open question: it depends on the nature of the data being collected and the model it is fitted to, and even the best methods require the researcher to make a number of assumptions about the population values of estimated parameters and/ or have initial data [24,25].

Throughout all studies, based on the estimated time to complete the full protocol, we set payment at the rate of £7.50 an hour.

## 2.3. Informed consent

All participants were presented with an online information sheet and subsequently gave online informed consent. They could also leave the study at any time by closing their browser. This study was approved by the University College London Research Ethics Committee (approval number 5253/001).

## 2.4. General analysis

The specific approach in each study is detailed under the study-specific subheadings below. However, there were some consistent approaches across studies: we tested for internal reliability and validity using Cronbach's alpha and omega, and examining inter-item and item-total correlations; we performed exploratory factor analyses in Studies 1 and 2, then confirmatory factor analyses in subsequent Studies 3, 4 and 5. These methods are further detailed in the electronic supplementary material, along with our policy for item reduction during scale development.

### 2.4.1. Study 1

#### 2.4.1.1. Scale
An initial scale with 31 items was used. Further details on how the items were developed can be found in the electronic supplementary material.

#### 2.4.1.2. Participants
We tested this first version of the questionnaire on a sample of 117 participants.

### 2.4.2. Study 2

In this second study, we reduced the number of items in the scale, and simplified items which were considered too long or too complex. Details of item removal can be found in the electronic supplementary material.

#### 2.4.2.1. Scale
Based on item reduction in Study 1, we tested a version of the scale with 25 items.

#### 2.4.2.2. Participants
This iteration of the questionnaire was tested on 117 participants.

### 2.4.3. Study 3

All participants completed the final version of the Catastrophizing Questionnaire along with convergent and discriminant validity questionnaires (all of which showed acceptable internal consistency in our sample, table 3).

#### 2.4.3.1. Scale

One item from the second version of the questionnaire was removed, resulting in a final version of the Catastrophizing Questionnaire with 24 items.

#### 2.4.3.2. Short version of the Catastrophizing Questionnaire

We developed a short version of the Catastrophizing Questionnaire, as there are a number of circumstances in which researchers or clinicians might need a briefer scale which corresponds to the full version. This short version consisted of four items. Three of them were from the Catastrophizing Questionnaire and were chosen based on their high factor loading and relevance to catastrophizing. The fourth item, 'I turn minor issues into really big problems in my head' which was derived from the first version of the questionnaire, reflected a more general definition of catastrophizing. We analysed the relationship between this questionnaire and its full version using a Pearson's correlation, and also analysed the relationship between this questionnaire without the fourth item and the full Catastrophizing Questionnaire.

#### 2.4.3.3. Participants

Five hundred participants provided good quality data in Study 3 (for additional data quality assurance checks, see electronic supplementary material).

#### 2.4.3.4. Convergent and discriminant measures

We collected data on a number of additional self-report questionnaires, in order to assess convergent and discriminant validity. We define construct validity here as positive associations between two measures of the same construct (i.e. two different questionnaires measuring the same thing), and define convergent validity as positive (but not large enough to indicate redundancy) associations between two measures of different constructs which are theoretically hypothesized to be related. We define discriminant validity as a lack of significant positive association between measures of constructs that are not hypothesized to be related to catastrophizing, but are related to mental health—such as anhedonia, schizotypy and alcohol use disorder. These measures are itemized below.

#### 2.4.3.5. Construct validity measures

*Cognitive Distortion Scale: Catastrophizing Subscale.* The Cognitive Distortion Scale (CDS) conceived by Covin *et al*. [20] contains a two-item catastrophizing subscale. This subscale was used for comparison with our Catastrophizing Questionnaire to assess construct validity.

#### 2.4.3.6. Convergent measures

*Anxiety.* Catastrophizing is believed to relate to psychiatric diagnoses such as anxiety and depression, and as such should be related to scores on questionnaires measuring symptoms of these disorders. Therefore, we included the Generalized Anxiety Disorder Assessment (GAD-7), which measures symptoms of generalized anxiety disorder over the previous two weeks [26]. We also included the Trait Anxiety Subscale of the State-Trait Anxiety Inventory (STAI-T) which measures anxiety level as an enduring personality trait, and focuses on anxiety in general rather than on a specific anxiety disorder [27].

   *Depression.* For the same reasons as above, we included the Patient Health Questionnaire (PHQ-9) which is a depression scale that asks about symptoms over the preceding two weeks [28,29]. As we were testing participants online and could not assess the risk of harm in person, we removed the question about suicidal ideation [30].

   *Worry.* Worry is an important feature of anxiety disorders that may also be related to catastrophizing. Thus, we included the Penn State Worry Questionnaire (PSWQ), which measures traits of worry independently from anxiety [31]. It examines the excessiveness, generality and uncontrollability of worry.

*Rumination.* Rumination is related to depression, and we thus hypothesized it would also correlate with catastrophizing. However, it is thought to be separable from depression, so in addition to the PHQ-9, we included the Rumination Response Scale (RRS) which measures two aspects of rumination: brooding and reflective pondering [32].

### 2.4.3.7. Discriminant measures

*Experience of pleasure.* We expected the experience of pleasure to be orthogonal to catastrophizing, as it implies a positive conception of the anticipation of pleasure and enjoying the present moment. Therefore, we included the Temporal Experience of Pleasure Scale—Anticipatory (TEPS-ANT) and Consummatory (TEPS-CON) subscales, which assess pleasure experienced during anticipation of reward and on attaining rewards, respectively [33].

### 2.4.3.8. Analysis of convergent and discriminant measures

We performed Pearson's correlations between scores on different questionnaires to assess the magnitudes of relationships between the putative latent constructs. To assess the extent to which the convergent measures examine separable constructs, we performed an exploratory factor analysis (EFA) including all these items, and hypothesized that the items from the Catastrophizing Questionnaire would load onto a separate factor to other items. We chose to use an EFA as a model-free way of attempting to characterize the relationships between individual items from all questionnaires—rather than assuming that each questionnaire is separate, we wanted to test whether the Catastrophizing Questionnaire items coupled together with other questionnaires, or whether they loaded separately onto a factor of their own. To assess discriminant validity, we also used the heterotrait–monotrait ratio of correlations (HTMT), which is a particularly robust measure of discriminant validity [34], for which statistic a value of less than 0.85 can be considered discriminant.

We also present an exploratory CFA analysis, performed using lavaan, in the electronic supplementary material, in which each 'convergent' questionnaire was mapped to a latent factor, such that each individual item within that questionnaire became an indicator for that latent factor, and subsequently estimated the covariances between each latent variable. We anticipated that this would result in heightened correlations between questionnaires, due to the disattenuation for measurement error for each indicator. The aim of this analysis was to assess whether the anticipated heightened correlations between items were specifically increased for correlations with the Catastrophizing Questionnaire, which might have indicated redundancy with the other questionnaire measures we used.

### 2.4.3.9. Self-reported medication, psychiatric diagnoses and catastrophizing: incremental validity

We also examined participants' responses to questions on their mental health and their medication history. We performed binomial logistic regression analyses to establish the incremental validity gained in predicting self-reported diagnostic status by using the Catastrophizing Questionnaire over two questionnaires which are commonly obtained in primary care settings in the UK—the GAD-7 and PHQ-9. A null model predicting the self-report diagnosis status of each participant using GAD-7 and PHQ-9 was compared with a model of interest which included the GAD-7, PHQ-9 and Catastrophizing Questionnaire scores. Similarly, we also performed multinomial logistic regression analyses on the self-reported medication status of each participant. We hypothesized that the Catastrophizing Questionnaire would show incremental validity over these questionnaires in predicting both self-report medication usage and diagnostic status.

### 2.4.4. Study 4

A number of participants (100) who had completed Study 3 repeated the final 24-item version of the Catastrophizing Questionnaire two months later. We did not systematically select these participants—the study was placed online and open to all participants who had completed Study 3, and recruitment proceeded in a 'first-come, first-served' manner.

### 2.4.4.1. Participants

This study was completed by an unselected, first-come, first-served group of 100 participants who had completed Study 3.

**Table 1.** Reliability and validity fit indices for the Catastrophizing Questionnaire. Internal consistency is measured by Cronbach's alpha, omega and inter-item correlations. Item-total correlations are used to assess discriminability of items—i.e. the extent to which they distinguish between high and low scorers on a task. While adequate values for these measures depend on the application, the nature of the construct and other factors, some suggested values for acceptability are given to aid interpretation. 0.8 (for basic or applied research) is considered to be an acceptable value for Cronbach's alpha, with 0.9 or above recommended for applied work that is guiding decision-making [35,36]. Similar values are recommended for omega. Mean inter-item correlations should be between 0.15 and 0.5 [35]. Item-total correlations should be over 0.3 or 0.4 [37].

| study | questionnaire version | Cronbach's alpha | omega | average inter-item correlation | average item-total correlation |
|---|---|---|---|---|---|
| Study 1 | 31 item version | 0.95 | 0.96 | 0.40 | 0.65 |
| Study 2 | 25 item version | 0.93 | 0.94 | 0.36 | 0.62 |
| Study 3 | 24 item version | 0.94 | 0.95 | 0.41 | 0.66 |
| Study 4 | 24 item version | 0.94 | 0.95 | 0.42 | 0.66 |
| Study 5 | 24 item version | 0.95 | 0.95 | 0.43 | 0.67 |

### 2.4.4.2. Test–retest reliability

We firstly calculated the Pearson's correlation coefficient between participants' scores in Studies 3 and 4. We also calculated the intraclass correlation coefficient (ICC). More details on the ICC can be found in the electronic supplementary material.

### 2.4.5. Study 5

A number of participants (264) who had completed Study 3 repeated the final 24-item version of the Catastrophizing Questionnaire 10 months later. They also completed two additional questionnaires to measure discriminant validity, and repeated the PHQ-9, GAD-7, PSWQ and RRS.

### 2.4.5.1. Participants

This study was completed by a subset of participants who had completed Study 3. As in Study 4, we did not systematically select these participants.

### 2.4.5.2. Discriminant measures

We collected data on two additional questionnaires: the Alcohol Use Disorder Identification Test (AUDIT), and the Short Scales for Measuring Schizotypy, both of which we did not expect to be highly related to catastrophizing, particularly over-and-above general psychiatric distress. More details on these can be found in the electronic supplementary material.

# 3. Results

All data and analyses are available online as a fully reproducible workbook here: https://doi.org/10.17605/OSF.IO/CRFUW.

## 3.1. Study 1

### 3.1.1. Reliability and validity

Cronbach's alpha, omega and the inter-item and inter-total correlations are presented in table 1, and all distributions of inter-item correlations are presented in the electronic supplementary material. All fit indices show initial evidence that the questionnaire is reliable and valid [35].

### 3.1.2. Exploratory factor analysis

The Kaiser–Meyer–Olkin (KMO) value was 0.92, and Bartlett's test of sphericity was significant, so our requirements for conducting an exploratory factor analysis were met. Parallel analysis indicated a three-factor structure. EFA fit indices are reported for all studies in the electronic supplementary material.

**Table 2.** Fit indices for confirmatory factor analysis solutions for the Catastrophizing Questionnaire.

| study | description | $\chi^2$ (d.f.)[a] | CFI[b] | TLI[c] | RMSEA[d] | SRMR[e] |
|---|---|---|---|---|---|---|
| Study 3 | CFA one-factor model | 1033 (252) | 0.87 | 0.86 | 0.08 | 0.05 |
| Study 4 | CFA one-factor model | 433 (252) | 0.85 | 0.84 | 0.09 | 0.07 |
| Study 5 | CFA one-factor model | 677 (252) | 0.87 | 0.86 | 0.08 | 0.05 |

[a]$\chi^2$ likelihood test.
[b]Comparative fit index.
[c]Tucker–Lewis Index (good fit indicated by close to 0.95).
[d]Root mean square error of approximation (good fit indicated by less than 0.08).
[e]Standardized root mean square residual (good fit indicated by less than 0.08).

## 3.2. Study 2

### 3.2.1. Reliability and validity

This version of the Catastrophizing Questionnaire also showed good internal consistency. Cronbach's alpha, omega, and the inter-item and inter-total correlations are presented in table 1.

### 3.2.2. Exploratory factor analysis

The KMO value was 0.9, and Bartlett's test of sphericity was significant, so our requirements for conducting an exploratory factor analysis were met. We performed an EFA in the same way as in the first study. Parallel analysis did not indicate conclusively how many factors should be considered, and solutions indicating one or two factor structures were both produced (parallel analysis, by generating 'random' datasets, is by definition stochastic: in the open code for this paper, we use two different seeds to demonstrate these different solutions, which the interested reader can inspect further—https://doi.org/10.17605/OSF.IO/CRFUW). However, the eigenvalue for the second factor was less than 1, indicating that the putative second factor is not stable, so we assumed a single-factor structure was the best fit to the data. The fit indices for the EFA are displayed in the electronic supplementary material.

## 3.3. Study 3

### 3.3.1. Reliability and validity

This final version of the Catastrophizing Questionnaire showed internal consistency. Cronbach's alpha, omega, and the inter-item and inter-total correlations are presented in table 1.

### 3.3.2. Confirmatory factor analysis

We ran a CFA specifying a one-factor model. This model had a $\chi^2_{252} = 1034$, SRMR of 0.05, a CFI of 0.87, a TLI of 0.86 and an RMSEA of 0.08 [CI: 0.074–0.084]. These fit indices are reported in table 2.

### 3.3.3. Short version of the Catastrophizing Questionnaire

The Catastrophizing Questionnaire had a strong correlation with its short version ($r_{498} = 0.81$, $p < 0.001$, figure 1b), which indicates that this could be used as a brief measure when necessary (although the correlation is not equal to 1, indicating that the full version contains more information). We include one item in the short version that did not overlap with the final version of the Catastrophizing Questionnaire. When this is omitted, the correlation between full and short version scores decreases ($r_{498} = 0.77$, $p < 0.001$). Notably, the short version is four times faster to complete than the full version: with participants in Study 3 taking a mean of 1 min and 59 s to respond to the full version, and a mean of 26 s to respond to the short version.

### 3.3.4. Convergent and discriminant validity

Internal consistency indices for all questionnaires used to test convergence are presented in table 3.

We ran an EFA with the Catastrophizing Questionnaire and the questionnaires used for convergent validity in order to determine the underlying factor structure that best represents the individual items (97

**Table 3.** Internal consistency indices for all measures used.

| measure | Cronbach's alpha | omega |
|---|---|---|
| GAD-7[a] | 0.90 | 0.93 |
| PHQ-9[b] | 0.89 | 0.91 |
| PSWQ[c] | 0.94 | 0.95 |
| RRS[d] | 0.94 | 0.95 |
| STAI-T[e] | 0.94 | 0.95 |
| TEPS-ANT[f] | 0.77 | 0.82 |
| TEPS-CON[g] | 0.68 | 0.77 |
| CDS-Catastrophizing[h] | 0.70 | N/A[i] |
| short version (4 items)[j] | 0.81 | 0.83 |
| AUDIT[k] | 0.88 | 0.92 |
| SS-UE[l] | 0.75 | 0.79 |
| SS-CD[m] | 0.83 | 0.86 |
| SS-IA[n] | 0.63 | 0.72 |
| SS-IN[o] | 0.55 | 0.62 |

[a]Generalized Anxiety Disorder Assessment.
[b]Personal Health Questionnaire.
[c]Penn-State Worry Questionnaire.
[d]Ruminative Responses Scale.
[e]Spielberger State-Trait Anxiety Inventory—Trait Subscale.
[f]Temporal Experience of Pleasure Scale, Anticipatory Subscale.
[g]Temporal Experience of Pleasure Scale, Consummatory Subscale.
[h]Cognitive Distortion Scale: Catastrophizing Subscale.
[i]There were too few items ($n = 2$) to calculate omega in the CDS-Catastrophizing scale.
[j]Short version of the full novel Catastrophizing Questionnaire reported in this paper.
[k]Alcohol Use Disorders Identification Test.
[l]Short Scales for Measuring Schizotypy—Unusual Experiences subscale.
[m]Short Scales for Measuring Schizotypy—Cognitive Disorganization subscale.
[n]Short Scales for Measuring Schizotypy—Introvertive Anhedonia subscale.
[o]Short Scales for Measuring Schizotypy—Impulsive Non-conformity subscale.

in total). The results, analysed using parallel analysis, suggested a seven-factor model. All fit indices are presented in the electronic supplementary material. Factor loadings are presented in a heatmap (figure 1*a*) which highlights that the items from the Catastrophizing Questionnaire heavily load onto one factor that is specific to catastrophizing. Therefore, these results suggest that catastrophizing may be a construct that is independent from worry, rumination, and low mood and anxiety, at least in our data.

Correlations between the Catastrophizing Questionnaire and convergent and discriminant measures were performed using the Pearson correlation test (figure 1*b*). When the same analysis was performed in a structural equation modelling framework, in order to disattenuate for measurement error, all the correlations increased, but this was not specific to correlations with the Catastrophizing Questionnaire alone (electronic supplementary material, figure S7). This provides further evidence for convergent validity without redundancy. The Catastrophizing Questionnaire had a moderate correlation with the CDS-Catastrophizing subscale ($r_{498} = 0.62$, $p < 0.001$), confirming its construct validity.

The Catastrophizing Questionnaire was more strongly associated with scores on the GAD-7 ($r_{498} = 0.70$, $p < 0.001$), STAI-T ($r_{498} = 0.78$, $p < 0.001$), PHQ-9 ($r_{498} = 0.67$, $p < 0.001$), PSWQ ($r_{498} = 0.72$, $p < 0.001$) and RRS ($r_{498} = 0.75$, $p < 0.001$) than with scores on the TEPS-ANT ($r_{498} = -0.17$, $p < 0.001$) and TEPS-CON ($r_{498} = -0.1$, $p = 0.02$), indicating that catastrophizing is more associated with measures of anxiety, depression, worry and rumination than with measures of the experience of pleasure or anhedonia. Furthermore, all $HTMT_{0.85}$ values for a model including the TEPS-ANT, TEPS-CON and Catastrophizing Questionnaire indicated discriminant validity (TEPS-ANT = 0.326, TEPS-CON = 0.182).

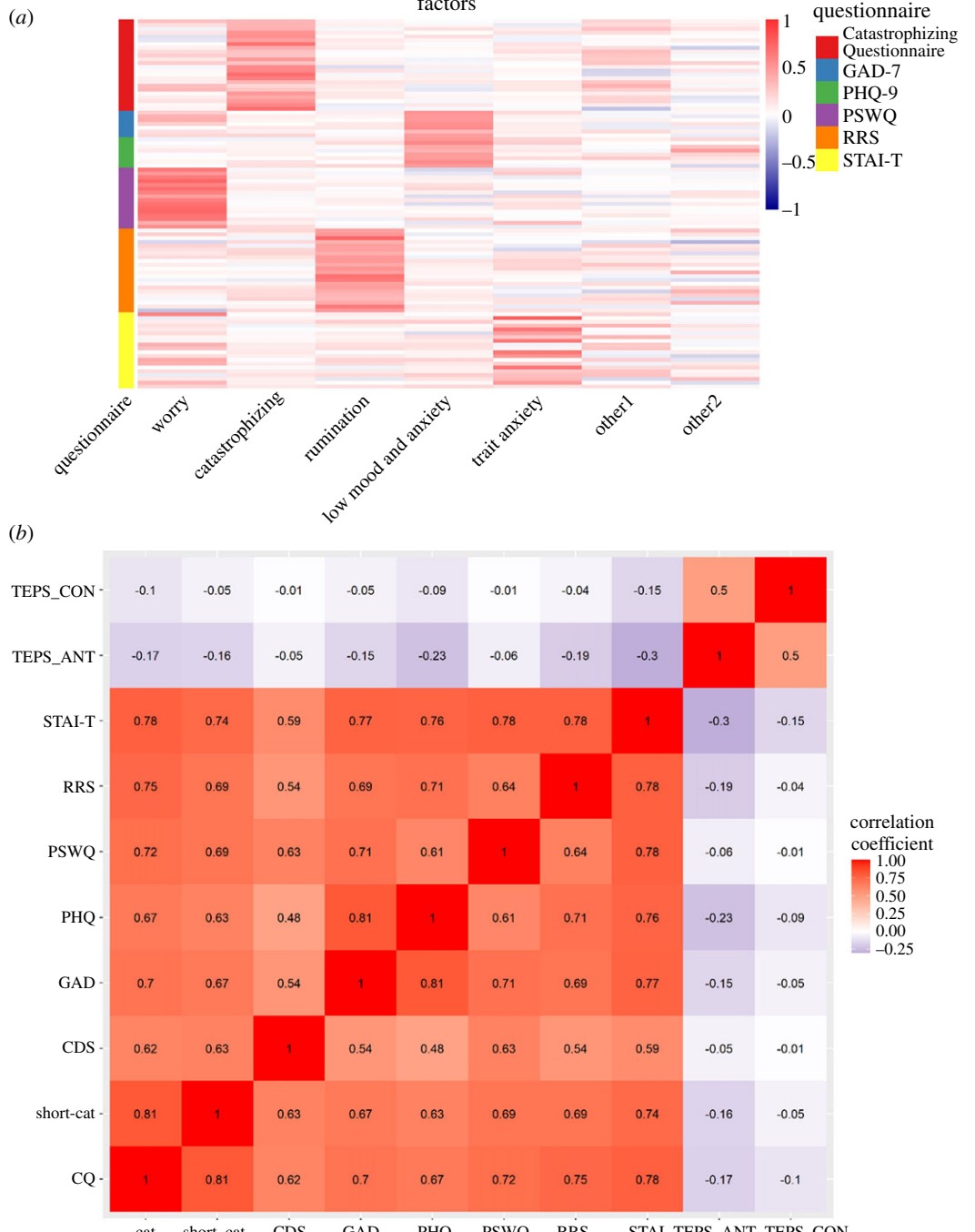

**Figure 1.** (*a*) Heatmap displaying factor loadings from the exploratory factor analysis of the Catastrophizing Questionnaire and all the scales used for convergent validity. (*b*) Correlation plot displaying the correlations between the Catastrophizing Questionnaire and all scales used as discriminant and convergent measures in Study 3. *Note*. CDS, CDS-Catastrophizing Subscale; GAD-7, Generalized Anxiety Disorder Assessment; PHQ-9, Personal Health Questionnaire; PSWQ, Penn State Worry Questionnaire; RRS, Ruminative Responses Scale; STAI-T, Spielberger State-Trait Anxiety Inventory—Trait Subscale; TEPS-CON, Temporal Experience of Pleasure Scale, Consummatory Subscale; TEPS-ANT, Temporal Experience of Pleasure Scale, Anticipatory Subscale; Short-cat, Short version of the Catastrophizing Questionnaire, CQ, Catastrophizing Questionnaire.

### 3.3.5. Self-reported medication, psychiatric diagnoses and catastrophizing: incremental validity

We tested incremental validity by seeing whether the Catastrophizing Questionnaire could predict (i) self-reported diagnostic status and (ii) self-reported medication status significantly better than using the PHQ-9 and GAD-7, which are in common clinical use within mental health services in the UK. A comparison of two logistic regression models, one including the PHQ-9 and GAD-7, and a nested model also including scores on the Catastrophizing Questionnaire, indicated that the Catastrophizing

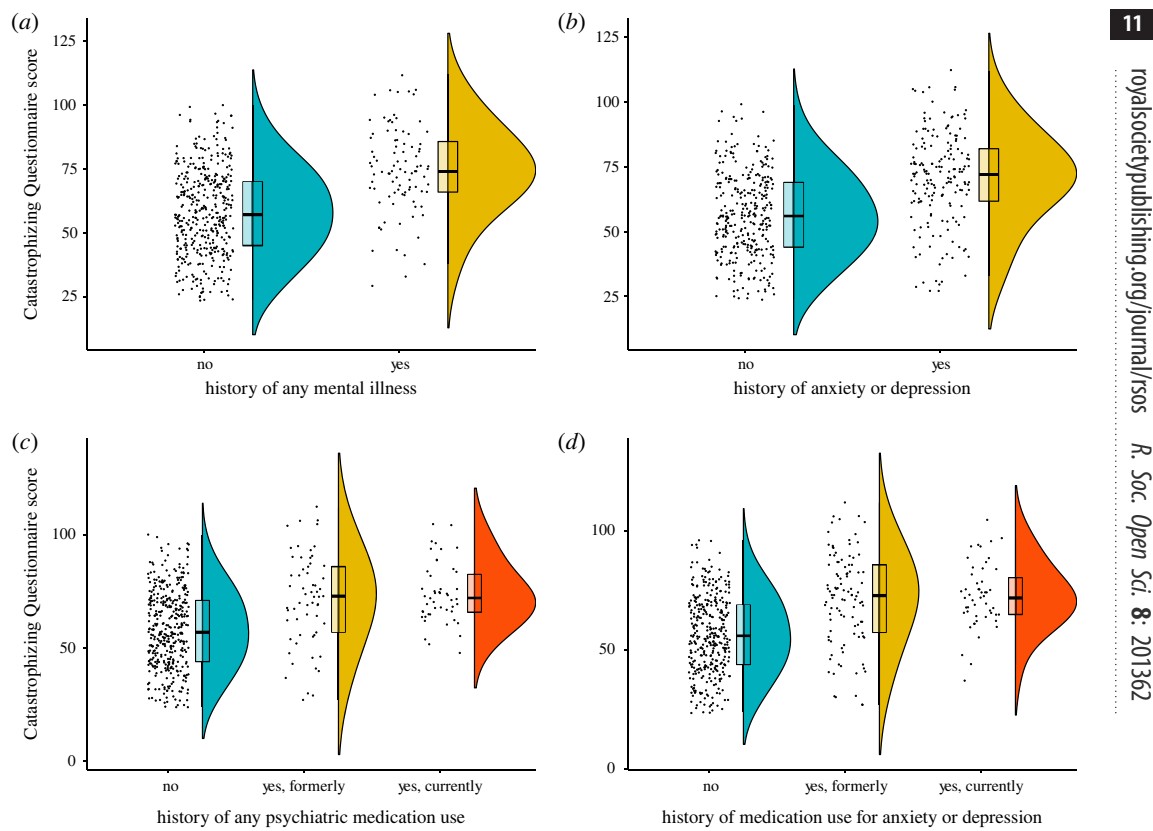

**Figure 2.** Raincloud plots displaying the means and standard deviations on the Catastrophizing Questionnaire from Study 3 separated by response to self-reported psychiatric diagnosis/medication questions. (*a*) Participants with and without a self-reported psychiatric diagnosis. (*b*) Participants with and without a self-reported diagnosis of anxiety or depression. (*c*) Participants who report currently taking psychiatric medication, who report taking these medications in the past, and participants who report that they have never taken these medications. (*d*) Participants who report that they are currently taking medication for anxiety or depression, who report that they have taken medication for anxiety or depression in the past, and participants who report that they have never taken any psychiatric medication for anxiety or depression (***$p < 0.001$).

Questionnaire was able to predict self-reported psychiatric diagnosis significantly better than using the PHQ-9 and GAD-7 alone ($\chi^2_{(1,N=500)} = 14.53$, $p = 0.00014$). This was also true when the dependent variable was self-reported (current or former) diagnosis of anxiety or depression specifically ($\chi^2_{(1,N=500)} = 9.74$, $p = 0.0018$). Analysis of self-reported psychiatric medication in a multinomial logistic regression also indicated incremental validity ($\chi^2_{(2,N=500)} = 12.99$, $p = 0.0015$). Finally, incremental validity was also shown when the dependent variable was self-reported medication for anxiety and/or depression more specifically ($\chi^2_{(2,N=500)} = 18.43$, $p = 0.0001$). The means and standard deviations on the Catastrophizing Questionnaire by self-reported diagnosis and psychiatric medication status are shown in electronic supplementary material, table S3 and in figure 2.

## 3.4. Study 4

An unselected subset of participants who completed Study 3 (100 participants) repeated the Catastrophizing Questionnaire two months later.

### 3.4.1. Reliability and validity

Replicating results from the previous three studies, the Catastrophizing Scale was internally consistent in the fourth study. Cronbach's alpha, omega, and the inter-item and inter-total correlations are presented in table 1.

### 3.4.2. Test–retest reliability

The retest coefficient was high ($r_{98} = 0.78$, $p < 0.001$) confirming the reliability of the scale. The ICC(A,1) was 0.774 [CI: 0.682, 0.842], which was significantly different from 0 ($F_{99,99.7} = 7.91$, $p < 0.001$)

and the ICC(C,1) was 0.776 [CI: 0.684, 0.843], which was also significantly different from 0 ($F_{99,99} = 7.91$, $p < 0.001$).

### 3.4.3. Confirmatory factor analysis

A one-factor model was an acceptable fit to the data. See table 2 for fit indices.

## 3.5. Study 5

### 3.5.1. Reliability and validity

Replicating results from the previous four studies, the Catastrophizing Scale was internally consistent in the fifth study. Cronbach's alpha, omega, and the inter-item and inter-total correlations are presented in table 1. It is notable that Cronbach's alpha and omega were both relatively low for some of the subscales of the Short Scales for Measuring Schizotypy.

### 3.5.2. Confirmatory factor analysis

A one-factor model was an acceptable fit to the data. See table 2 for fit indices.

### 3.5.3. Convergent and discriminant validity

Internal consistency indices for all items are presented in table 3. Scores on the Catastrophizing Questionnaire showed a small but significant correlation with scores on the AUDIT ($r_{262} = 0.27$, $p < 0.001$). The $HTMT_{0.85}$ was 0.315, below the 0.85 cut-off to indicate discriminant validity. Catastrophizing Questionnaire scores showed a range of different correlations with the four facets on the Short Scales for Measuring Schizotypy (Unusual Experiences: $r_{262} = 0.47$, $p < 0.001$; Cognitive Disorganization: $r_{262} = 0.63$ $p < 0.001$; Introvertive Anhedonia: $r_{262} = 0.34$, $p < 0.001$; Impulsive Non-conformity: $r_{262} = 0.46$, $p < 0.001$. All $HTMT_{0.85}$ values indicated discriminant validity (UE = 0.555, CD = 0.705, IA = 0.508, IN = 0.650).

In summary, while these discriminant measures all showed positive correlations with catastrophizing, many of the correlations are small to moderate, and all of the HTMT values indicated discriminant validity.

## 4. Discussion

We have developed the Catastrophizing Questionnaire to provide a comprehensive self-report measure of a cognitive process that is related to diagnoses such as anxiety and depression but which, until recently, could only be measured in the context of pain. The questionnaire demonstrates high reliability and internal consistency and measures a unitary construct that is separable from anxiety, depression and worry.

In Studies 1 and 2, we refined an initial version of a novel Catastrophizing Questionnaire and examined the construct properties using exploratory factor analysis. In Studies 3–5, we were able to show acceptable fit for a single-factor structure using an out-of-sample CFA, and demonstrated that all 24 items from the novel questionnaire loaded highly onto this factor, indicating that catastrophizing can be considered a unitary construct. This finding is a little surprising, given the definition of catastrophizing that we propose above, which includes two components—the overestimation of the probability of rare negative events, and the overestimation of the magnitude or severity of negative events. We did not specifically test this hypothesis using a CFA in our data, and it would not be wise to do so after performing an EFA [38], but this may be a fruitful avenue for future work. It is possible that a wider item pool, with more items specifically referring to each separate component, would have shown such a two-factor structure. Study 4 showed that test–retest reliability for this construct was high (ICC(A,1) = 0.77) over a two-month period.

We demonstrated that the Catastrophizing Questionnaire showed convergent and discriminant validity in Studies 3 and 5. Catastrophizing was positively related to depression, anxiety, rumination and worry, but discriminant validity was demonstrated using measures of the experience of pleasure/ anhedonia, alcohol use disorders and schizotypy.

Critically, however, despite some convergence, this construct is also *dissociable* from other psychiatric constructs. The multi-questionnaire EFA in Study 3 (including the Catastrophizing Questionnaire, PHQ-9, GAD-7, RRS, PSWQ and the STAI-T) indicated that catastrophizing is independent from convergent constructs such as anxiety and low mood, worry and rumination. Specifically, the items from the Catastrophizing Questionnaire loaded onto a single factor, separate from the items of the other questionnaires. This work could be further extended, by also testing whether catastrophizing is independent from other cognitive distortions such as black-and-white thinking or overgeneralization. However, the findings reported thus far allow us to begin to address one of the important questions in the study of catastrophizing: whether catastrophizing should be considered as a construct independent from psychiatric diagnoses such as anxiety and depression. Our results indicate that catastrophizing is a unitary construct, independent of anxiety, depression, rumination or worry. Therefore, we suggest that catastrophizing is not just an epiphenomenon or a straightforward consequence of anxiety and depression, but may be a separable construct with at least partially independent aetiology (although, notably, factor analytic methods may not always accurately capture the underlying factor structure of a construct [39]). A personalized medicine approach could thus be brought to bear: those with high levels of catastrophizing could receive therapy (such as decatastrophizing) targeted at this cognitive process. Future work should, therefore, investigate whether decatastrophizing therapy in CBT is able to specifically reduce catastrophizing (as measured using our Catastrophizing Questionnaire), as no study has yet assessed the mechanisms by which decatastrophizing promotes improvement of psychiatric or mood symptoms.

Importantly, results on the Catastrophizing Questionnaire were also predictive of self-reported psychiatric diagnosis and medication history, over-and-above commonly used clinical scales (the PHQ-9 and GAD-7). Thus, we have shown that this construct is relevant to the field of psychiatry, and shows incremental validity over other measures. We believe, therefore, that our measure of catastrophizing captures an important additional risk factor for diagnosis that is not present in existing scales. From a clinical perspective, we believe that having the ability to measure general catastrophizing will facilitate the assessment of the efficacy and mechanism-of-action of de-catastrophizing approaches, and, furthermore, items on the scale may potentially provide patient-specific directions for de-catastrophizing during therapy.

The items on the Catastrophizing Questionnaire are related to items used in pain catastrophizing measures—indeed, we used pain catastrophizing scales as inspiration for our bank of items. However, many of the items included in pain catastrophizing scales either assume a current experience of pain—for example 'there's nothing I can do to reduce the intensity of the pain'—or at least assume a current difficult experience—for example 'I worry all the time about whether it will end'. Our questionnaire should be applicable to the general population, as well as psychiatric populations, by not presupposing these points.

## 5. Limitations

Despite its strengths, the Catastrophizing Questionnaire has several limitations to consider. First, we based the definition of catastrophizing on our understanding of it, seeking additional guidance from the research literature in the context of pain and from experts in the field. However, there is not yet consensus on a precise definition of catastrophizing outside of the context of pain, meaning that subjective judgement was required when developing the items.

Relatedly, we did not include pain catastrophizing measures in our study of convergent validity, which may have given us valuable additional information. Our justification for this was that many of the questions in pain catastrophizing scales explicitly discuss a current pain experience, making them inappropriate to administer to a general sample of the population.

Second, catastrophizing may be rooted in one or more specific contexts for each individual. We aimed at including this factor in the Catastrophizing Questionnaire by having examples of catastrophizing for each context. We provided specific situations which may not apply to all individuals. Nevertheless, the examples provided were kept as general as possible in all contexts.

Third, our use of rules of thumb for determining sample size is suboptimal. Notably, however, Monte Carlo simulation studies in general suggest that while our first two studies are likely to be underpowered for obtaining a factor structure, our third study, on which we base the majority of our conclusions, is likely to be sufficiently powered even if there are many weakly determined factors [40]. Relatedly,

future research should provide further validation of the Catastrophizing Questionnaire in a larger population in order to confirm the factor model.

Fourth, we did not have an inbuilt method such as an attention check to detect careless responding: though, notably, in Experiment 3 (on which we base the majority of our conclusions), we manually check data from individuals who repeatedly gave the same responses or gave different responses to repeated questions.

Fifth, it should be noted that for Studies 3, 4 and 5, testing the final scale, fit indices did not universally indicate good fit of a single-factor model. However, fit values were still close to the requirements; thus, our fit indices indicate a reliable analysis.

Sixth, as with any questionnaire, it takes some time to complete (around 2 min). To this end, we also created a short version of the Catastrophizing Questionnaire. The challenge in developing a short form of an existing questionnaire is to maintain the level of information while significantly reducing the length of the scale. A further limitation to this study is that we could have based the development of the short-form questionnaire on Item Response Theory analysis which enables the identification of items yielding maximum information [35]. Indeed, the development of the short-form questionnaire was mostly based on theoretical considerations. Additionally, one of the four items constituting the short-form questionnaire was not in the Catastrophizing Questionnaire from Study 3 and was derived from the Catastrophizing Questionnaire of Study 1. This procedure may seem unusual for a short-form questionnaire which is meant to contain information from the original version. The item that we included was originally part of the initial item bank, but was removed from later versions of the questionnaire as other items were more specific. However, when developing a short form, we theorized that more general items would be more valuable, given the limited data collected using a short form of any questionnaire. Nevertheless, this short version demonstrated good internal consistency and correlated highly with the full version, indicating that it could be used as a brief measure (around 30 s for completion) where necessary. Researchers could choose to use the 3-item short form, which includes only items present in the final version of the Catastrophizing Questionnaire, though removing the non-overlapping item reduces the correlation between scores on the short and full version.

Finally, although we concluded from the EFA in Study 3 that catastrophizing may be considered independent from other closely related constructs, it may be argued that the EFA was detecting patterns of responding to similarly phrased items rather than the true structure underlying these latent variables. However, many of the items in our questionnaire share phrasing with those from other questionnaires, in particular the RRS, but do not load onto the same factor, arguing against this interpretation.

## 6. Conclusion

Overall, our new Catastrophizing Questionnaire is the first self-report measure of the general cognitive process of catastrophizing, with strong evidence of reliability and validity across multiple studies. It offers a powerful tool for probing the nature and the progression over time of catastrophizing (and the effects of interventions), with the potential to advance understanding of its contribution to the development or maintenance of mental ill-health and distress. Given the rising interest in catastrophizing in CBT and as a potential risk factor for a range of psychiatric symptoms, the Catastrophizing Questionnaire may be useful to many clinicians and researchers.

Ethics. All participants were presented with an online information sheet and subsequently gave online informed consent. They could also leave the study at any time by closing their browser. This study was approved by the University College London Research Ethics Committee (approval number 5253/001).

Data accessibility. All data and analyses are available online as a fully reproducible R notebook here: https://doi.org/10.17605/OSF.IO/CRFUW.

Authors' contributions. A.C.P. and O.J.R. conceived the project. A.C.P. and J.R.S. collected and analysed the data, with supervision by O.J.R. All authors drafted the manuscript and revised it. All authors have approved the final article.

Competing interests. O.J.R.'s MRC senior fellowship is partially in collaboration with Cambridge Cognition (who plan to provide in-kind contribution) and he is running an investigator-initiated trial with medication donated by Lundbeck (escitalopram and placebo, no financial contribution). He also holds an MRC-Proximity to discovery award with Roche (who provide in-kind contributions and have sponsored travel for A.C.P.) regarding work on heart rate variability and anxiety. He has also completed consultancy work on affective bias modification for Peak and online CBT for IESO digital health. J.R.S. and A.C.P. declare no other conflicts of interest. Neither the MRC or Wellcome Trust (who provided funding) had any role in the study design, collection, analysis or interpretation of data.

Funding. This work was supported by a Medical Research Council senior non-clinical fellowship (MR/R020817/1) to O.J.R.; and a Biomedical Vacation Scholarship awarded to J.R.S. by the Wellcome Trust (218411_Z_19_Z_UCL). Acknowledgements. We are also grateful for helpful conversations with Professors Paul Burgess, Roz Shafran, John Cape and Vaughan Bell. Finally, the authors would like to thank the other members of the Neuroscience and Mental Health Group for their help with brainstorming and piloting.

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
