## [Reviewer comments · Royal Society Open Science]

Review History

RSOS-201362.R0 (Original submission)

Review form: Reviewer 1 (Katherine Young)

Is the manuscript scientifically sound in its present form?

Yes

Are the interpretations and conclusions justified by the results?

Yes

Is the language acceptable?

Yes

Do you have any ethical concerns with this paper?

No

Have you any concerns about statistical analyses in this paper?

No

Recommendation?

Accept with minor revision (please list in comments)

Comments to the Author(s)

Overall comment:

The authors present a comprehensive set of analyses assessing the psychometric properties of a newly developed self-report measure of catastrophising. This is an impressive and detailed body of work in an area surprisingly lacking in extant measures. In particular, the authors are to be commended for their comprehensive analyses and their open science approach to their materials and code. This is a well-written paper and overall provides a good overview of the work conducted. However, there are a few areas requiring improvement – importantly regarding the rationale for the work presented and interpretation of findings. These are detailed below, along some more minor comments.

Major comments:

- While the introduction is largely clear and well-written, there appears to be a gap in the rationale for the current work. The authors state there are few existing measures and those that do exist focus on pain-related catastrophising. However, they go on to state that existing self-report questionnaires do have some limited items on catastrophising (p.5). More detail here would be useful. What are these scales? What are the limitations of these measures? Why is a full scale needed? (& how does the limited number of items used in previous work sit with the development of a 4-item short version in the current work?)
- Relatedly, the statement “Studies have looked at the item content of scales measuring catastrophising, depression and anxiety and found that they were markedly similar, although intended to measure distinct constructs (9).” requires further explanation. What is the reasoning to suggest that catastrophising is actually a separable construct if previous studies have demonstrated such similarities? Are you suggesting there’s a limitation in the operationalisation of catastrophising as a construct in this work? (or are these pain-related catastrophising measures?) Explaining these limitations more fully is important in justifying the rationale for the work presented here
- The inclusion of an item in the short version that is not included in the long version seems unusual and is not well justified or explained. Further detail on this decision would be useful.
- I am unsure of the conclusion that “catastrophising may be a construct that is independent from anxiety, worry, rumination, and depression, at least in our data.” On p.18, this refers to figure 1a which shows separation from other factors labelled ‘worry’, ‘rumination’, ‘low mood and anxiety’, ‘trait anxiety’ and ‘other’. I think the description here might best be limited to these specific variables (especially given the fairly high correlations between ‘anxiety’ (GAD-7) and ‘depression’ (PHQ-9) shown in Fig 1B). Similarly, on p.23 – this might be rephrased to more accurately reflect the findings.
- I find the phrasing ‘a unitary construct that is separable from anxiety, depression and worry’ to be somewhat confusing. Throughout, it is unclear if the authors consider catastrophising to be a symptom of anxiety or more of a ‘process variable’, a trait-level cognitive process which might contribute to development or maintenance of anxiety or depression (similar to intolerance of uncertainty, or anxiety sensitivity for example; Norton et al., 2005). These process variables would be considered at a different level to either diagnoses (anxiety, depression) or symptoms (e.g., worry). It might be anticipated that higher levels of catastrophising tendencies may manifest as more frequent worry (symptom) and contribute to anxiety or depression (diagnosis). Alternatively, if they consider catastrophising to be a specific

symptom that is not adequately measured in existing diagnostic questionnaires, this should be addressed in the manuscript. Clarity on the author's conceptualisation of catastrophising may then help to make some of the more ambiguous phrasing (such as that specified above) more concrete.

- Regarding discriminant validity – only the TEPS scores are specifically stated as showing discriminant validity (p.20). Was there a definition of 'convergent' validity? Do the authors consider the associations with GAD/STAI/PSWQ/RRS to be convergent or discriminant here? From the discussion of these findings, the authors consider catastrophising to be separable from anxiety/depression etc. (p.23), but also positively related to these measures (p.24). Again clarity on the types of variables involved might help improve the interpretability of these statements (e.g., catastrophising as a process variable related to anxiety and depression, but clearly loading onto a separate latent factor compared to worry/rumination etc. – or catastrophising as a separate symptom cluster??)
- Regarding the incremental validity findings. Do the authors have an interpretation for these findings? Why might a measure of catastrophising improve prediction of diagnosis and medication usage? Is this simply a sensitivity finding (i.e., with an additional measure you can capture more variance?) – or do the authors think there are aspects of catastrophising that are particularly strong indicators of psychiatric diagnoses and medication usage?

Minor comments:

Abstract:

- Might be helpful to describe catastrophising as a cognitive process in the first sentence (as in para 2 of the introduction)
- The final sentence states catastrophising is a psychiatric symptom – as mentioned above, clarity on symptom vs. process level variable would be useful

Methods:

- Power calculations are discussed in the limitations section, but not mentioned in the methods and don't appear to be in the supplementary materials?
- Quality assurance checks are mentioned for study 3, but not for studies 1 or 2 – were these included only for study 3? Is there a reason for this?
- The ordering of justification of the additional scales could be improved. After reading the entire paper it is clear what the rationale for including each measure is, but when these first appear in the methods, this is unclear. It might be prudent to review the ordering of this information – and at least describe in a brief, broad overview what the purpose of these scales is before they are mentioned in detail.
- Results of the Lavaan CFA in the supplementary materials are described in the methods section – these would be better placed in context in the results section
- Findings reported in the results section could use more context (e.g., p.15 – 'All indices are presented in Table 1'; it is not immediately apparent what this refers to)
- Table 1 – it would be useful to provide for the reader the definition of 'reliable' and 'valid' based on the values reported; adding a brief summary of the scale used (e.g., catastrophising scale – 24 item version) to the column currently listing 'study 1' etc. would be useful
- The supplementary materials reports that "in all our studies, the participants who gave us feedback generally judged the Catastrophising Questionnaire to be clear, relevant to their experiences, and easy to complete" – if this is to be reported, more detail on the types of questions and mean scores should be included to allow the reader to assess this statement for themselves

- The statement 'whether clinicians thought an item captured elements of catastrophising' requires further explanation (which clinicians - what expertise? How many? Was this a formal or informal process?)

Additional reference:

Norton, P. J., Sexton, K. A., Walker, J. R., & Ron Norton, G. (2005). Hierarchical model of vulnerabilities for anxiety: Replication and extension with a clinical sample. *Cognitive Behaviour Therapy*, 34(1), 50-63.

Review form: Reviewer 2

Is the manuscript scientifically sound in its present form?

Yes

Are the interpretations and conclusions justified by the results?

Yes

Is the language acceptable?

Yes

Do you have any ethical concerns with this paper?

No

Have you any concerns about statistical analyses in this paper?

No

Recommendation?

Accept with minor revision (please list in comments)

Comments to the Author(s)

See attached PDF (Appendix A).

Decision letter (RSOS-201362.R0)

Dear Dr Pike

On behalf of the Editors, we are pleased to inform you that your Manuscript RSOS-201362 "The development and psychometric properties of a self-report Catastrophising Questionnaire" has been accepted for publication in Royal Society Open Science subject to minor revision in accordance with the referees' reports. Please find the referees' comments along with any feedback from the Editors below my signature.

We invite you to respond to the comments and revise your manuscript. Below the referees' and Editors' comments (where applicable) we provide additional requirements. Final acceptance of

your manuscript is dependent on these requirements being met. We provide guidance below to help you prepare your revision.

Please submit your revised manuscript and required files (see below) no later than 7 days from today's (ie 20-Nov-2020) date. Note: the ScholarOne system will 'lock' if submission of the revision is attempted 7 or more days after the deadline. If you do not think you will be able to meet this deadline please contact the editorial office immediately.

on behalf of Dr Emma Hayiou-Thomas (Associate Editor) and Essi Viding (Subject Editor)
openscience@royalsociety.org

Associate Editor Comments to Author (Dr Emma Hayiou-Thomas):

Associate Editor: 1

Comments to the Author:

I am in full agreement with both reviewers that this is a strong paper, filling an important gap in the literature, using a systematic set of rigorous and well thought-through studies. The open-science approach you've taken was also highlighted as a notable strength. The reviewers have provided a set of thoughtful and constructive suggestions for further improvement of the manuscript. I'd particularly like to highlight two points: 1) the request for clarification of how catastrophising is conceptualised in your scale: which level of variable you see it as (symptom, cognitive process, or diagnosis; R1), and the question of unidimensionality, (R2). 2) a more explicit comparison of your scale with the small sets of comparable items used in other scales in the literature (e.g. in the context of pain catastrophising).

Reviewer comments to Author:

Reviewer: 1

Comments to the Author(s)

Overall comment:

The authors present a comprehensive set of analyses assessing the psychometric properties of a newly developed self-report measure of catastrophising. This is an impressive and detailed body of work in an area surprisingly lacking in extant measures. In particular, the authors are to be commended for their comprehensive analyses and their open science approach to their materials and code. This is a well-written paper and overall provides a good overview of the work conducted. However, there are a few areas requiring improvement – importantly regarding the

rationale for the work presented and interpretation of findings. These are detailed below, along with some more minor comments.

Major comments:

- While the introduction is largely clear and well-written, there appears to be a gap in the rationale for the current work. The authors state there are few existing measures and those that do exist focus on pain-related catastrophising. However, they go on to state that existing self-report questionnaires do have some limited items on catastrophising (p.5). More detail here would be useful. What are these scales? What are the limitations of these measures? Why is a full scale needed? (& how does the limited number of items used in previous work sit with the development of a 4-item short version in the current work?)

- Relatedly, the statement "Studies have looked at the item content of scales measuring catastrophising, depression and anxiety and found that they were markedly similar, although intended to measure distinct constructs (9)." requires further explanation. What is the reasoning to suggest that catastrophising is actually a separable construct if previous studies have demonstrated such similarities? Are you suggesting there's a limitation in the operationalisation of catastrophising as a construct in this work? (or are these pain-related catastrophising measures?) Explaining these limitations more fully is important in justifying the rationale for the work presented here

- The inclusion of an item in the short version that is not included in the long version seems unusual and is not well justified or explained. Further detail on this decision would be useful.

- I am unsure of the conclusion that "catastrophising may be a construct that is independent from anxiety, worry, rumination, and depression, at least in our data." On p.18, this refers to figure 1a which shows separation from other factors labelled 'worry', 'rumination', 'low mood and anxiety', 'trait anxiety' and 'other'. I think the description here might best be limited to these specific variables (especially given the fairly high correlations between 'anxiety' (GAD-7) and 'depression' (PHQ-9) shown in Fig 1B). Similarly, on p.23 - this might be rephrased to more accurately reflect the findings.

- I find the phrasing 'a unitary construct that is separable from anxiety, depression and worry' to be somewhat confusing. Throughout, it is unclear if the authors consider catastrophising to be a symptom of anxiety or more of a 'process variable', a trait-level cognitive process which might contribute to development or maintenance of anxiety or depression (similar to intolerance of uncertainty, or anxiety sensitivity for example; Norton et al., 2005). These process variables would be considered at a different level to either diagnoses (anxiety, depression) or symptoms (e.g., worry). It might be anticipated that higher levels of catastrophising tendencies may manifest as more frequent worry (symptom) and contribute to anxiety or depression (diagnosis). Alternatively, if they consider catastrophising to be a specific symptom that is not adequately measured in existing diagnostic questionnaires, this should be addressed in the manuscript. Clarity on the author's conceptualisation of catastrophising may then help to make some of the more ambiguous phrasing (such as that specified above) more concrete.

- Regarding discriminant validity - only the TEPS scores are specifically stated as showing discriminant validity (p.20). Was there a definition of 'convergent' validity? Do the authors consider the associations with GAD/STAI/PSWQ/RRS to be convergent or discriminant here? From the discussion of these findings, the authors consider catastrophising to be separable from anxiety/depression etc. (p.23), but also positively related to these measures (p.24). Again clarity on the types of variables involved might help improve the interpretability of these statements (e.g., catastrophising as a process variable related to anxiety and depression, but clearly loading

onto a separate latent factor compared to worry/rumination etc. – or catastrophising as a separate symptom cluster??)

- Regarding the incremental validity findings. Do the authors have an interpretation for these findings? Why might a measure of catastrophising improve prediction of diagnosis and medication usage? Is this simply a sensitivity finding (i.e., with an additional measure you can capture more variance?) – or do the authors think there are aspects of catastrophising that are particularly strong indicators of psychiatric diagnoses and medication usage?

Minor comments:

Abstract:

- Might be helpful to describe catastrophising as a cognitive process in the first sentence (as in para 2 of the introduction)
 - The final sentence states catastrophising is a psychiatric symptom – as mentioned above, clarity on symptom vs. process level variable would be useful

Methods:

- Power calculations are discussed in the limitations section, but not mentioned in the methods and don't appear to be in the supplementary materials?
 - Quality assurance checks are mentioned for study 3, but not for studies 1 or 2 – were these included only for study 3? Is there a reason for this?
 - The ordering of justification of the additional scales could be improved. After reading the entire paper it is clear what the rationale for including each measure is, but when these first appear in the methods, this is unclear. It might be prudent to review the ordering of this information – and at least describe in a brief, broad overview what the purpose of these scales is before they are mentioned in detail.
 - Results of the Lavaan CFA in the supplementary materials are described in the methods section – these would be better placed in context in the results section
 - Findings reported in the results section could use more context (e.g., p.15 – 'All indices are presented in Table 1'; it is not immediately apparent what this refers to)
 - Table 1 – it would be useful to provide for the reader the definition of 'reliable' and 'valid' based on the values reported; adding a brief summary of the scale used (e.g., catastrophising scale – 24 item version) to the column currently listing 'study 1' etc. would be useful
 - The supplementary materials reports that "in all our studies, the participants who gave us feedback generally judged the Catastrophising Questionnaire to be clear, relevant to their experiences, and easy to complete" – if this is to be reported, more detail on the types of questions and mean scores should be included to allow the reader to assess this statement for themselves
 - The statement 'whether clinicians thought an item captured elements of catastrophising' requires further explanation (which clinicians – what expertise? How many? Was this a formal or informal process?)

Additional reference:

Norton, P. J., Sexton, K. A., Walker, J. R., & Ron Norton, G. (2005). Hierarchical model of vulnerabilities for anxiety: Replication and extension with a clinical sample. *Cognitive Behaviour Therapy*, 34(1), 50-63.

Reviewer: 2

Comments to the Author(s)

See attached PDF

===PREPARING YOUR MANUSCRIPT===

===PREPARING YOUR REVISION IN SCHOLARONE===

- An individual file of each figure (EPS or print-quality PDF preferred [either format should be produced directly from original creation package], or original software format).
 - An editable file of each table (.doc, .docx, .xls, .xlsx, or .csv).
 - An editable file of all figure and table captions.
- Note: you may upload the figure, table, and caption files in a single Zip folder.
- Any electronic supplementary material (ESM).
 - If you are requesting a discretionary waiver for the article processing charge, the waiver form must be included at this step.
 - If you are providing image files for potential cover images, please upload these at this step, and inform the editorial office you have done so. You must hold the copyright to any image provided.
 - A copy of your point-by-point response to referees and Editors. This will expedite the preparation of your proof.

- Ensure that your data access statement meets the requirements at <https://royalsociety.org/journals/authors/author-guidelines/#data>. You should ensure that you cite the dataset in your reference list. If you have deposited data etc in the Dryad repository, please only include the 'For publication' link at this stage. You should remove the 'For review' link.
- If you are requesting an article processing charge waiver, you must select the relevant waiver option (if requesting a discretionary waiver, the form should have been uploaded at Step 3 'File upload' above).
- If you have uploaded ESM files, please ensure you follow the guidance at <https://royalsociety.org/journals/authors/author-guidelines/#supplementary-material> to include a suitable title and informative caption. An example of appropriate titling and captioning may be found at https://figshare.com/articles/Table_S2_from_Is_there_a_trade-off_between_peak_performance_and_performance_breadth_across_temperatures_for_aerobic_scope_in_teleost_fishes_/3843624.

Author's Response to Decision Letter for (RSOS-201362.R0)

See Appendix B.

Decision letter (RSOS-201362.R1)

Dear Dr Pike,

It is a pleasure to accept your manuscript entitled "The development and psychometric properties of a self-report Catastrophising Questionnaire" in its current form for publication in Royal Society Open Science. The comments from the Editors are included at the foot of this letter.

on behalf of Dr Emma Hayiou-Thomas (Associate Editor) and Essi Viding (Subject Editor)
openscience@royalsociety.org

Associate Editor Comments to Author (Dr Emma Hayiou-Thomas):

Thank you for your comprehensive and thoughtful response to the reviews. The clarifications you have added make for an even stronger paper, which I believe will make a valuable contribution to the literature.

Appendix A

Summary:

In the current study, the authors developed a self-report scale to measure catastrophizing, which is comprised of two facets including: predicting an objective catastrophe will occur to oneself (e.g., a heart attack, which others agree would be catastrophic), or having the subjective perception that a negative/unwanted outcome will be catastrophic (e.g., social rejection being perceived as catastrophic, when most other people would not assume so). In both cases, catastrophizing involves assuming that the worse will happen.

The authors chose to focus on catastrophizing due to its prominence in cognitive behavioral theories of anxiety, depression, and other forms of psychopathology (i.e. it is one of many “cognitive distortions” that is often present transdiagnostically). As they mention, “de-catastrophizing” is a key component of cognitive behavioral therapy, as catastrophizing is thought to play a causal role in the persistence of anxiety and avoidance. Importantly, there are no self-reports of catastrophizing outside of the context of pain catastrophizing, which is an important oversight given its clinical relevance.

Given the lack of self-reports available for measuring catastrophizing, the authors developed and validated their own pool of items to do so. They use multiple samples to refine their item pool, test for convergent and discriminant validity, and develop a short-form version of the full scale. The authors show that a 24-item full version and a 4-item short version scale both exhibit acceptable fit statistics (according to a single factor, factor analytic model).

Strengths:

I think that the authors chose a really important topic, and they identified a clear gap in the literature that I was surprised to learn that no previously existing measure was developed to fill. Catastrophizing is indeed a key component of cognitive behavioral theories and treatments of psychopathology, and the availability of a measure should be very useful for researchers and clinicians alike (particularly the short version in the case of clinical application). Additionally, the authors conduct a rather comprehensive set of analyses to show how their catastrophizing measure converges with other similar constructs and diverges from those that we would not expect it to correlate with. Finally, the authors make their code and self-report items available online, which is very beneficial to the research community.

Major concerns:

Overall, I think this was a really good study and it provides an important addition to the literature. I have a few concerns that I think should be addressed before publication of this manuscript, although none require substantial changes to the analyses or text. My most important concerns are as follows:

- 1) My first concern is regarding the unidimensionality of catastrophizing. For example, as described by the authors in the introduction and noted above in my summary, two facets

including: (a) predicting an objective catastrophe will occur to oneself; and (2) having the subjective perception that a negative/unwanted outcome will be catastrophic, fall under the general definition of catastrophizing. In clinical settings, these two facets are often discussed to reflect: (a) overestimation of the probability of negative events occurring; and (2) underestimation of ability to cope with a negative event that occurs, or overestimating how bad the negative event would actually be (e.g., Foa & Kozak, 1986; 10.1037/0033-2909.99.1.20; McNally & Foa, 1987; 10.1007/BF01183859). Although these are related, they are distinct in that one involves estimating event probabilities, and the other costs/values. I think some discussion of these ideas could be important to include in the manuscript. For example, would the authors expect to find multiple factors if they had a wider item pool to assess these different mechanisms to start with? Looking at the items that the authors included, it looks like they do cover both probability and value estimation/valuation. They could potentially test this idea using CFA, but I think it would at least be worth discussing in the context of their 1 factor model.

2) My second concern is partly due to the formatting/organization of the text, but also a conceptual concern. I had trouble following the rationale for why the specific scales in Table 3 (which appears before Tables 1 and 2 in the text, which partly explains my confusion) were included to test the convergent and discriminant validity of the new Catastrophizing Scale. The authors note that they include the PHQ-9 and GAD-7 scales due to their widespread use for measuring general anxiety and depression in clinical settings in the UK (which I think is a solid rationale). Also, their inclusion of rumination, trait anxiety, and worry related measures (i.e. the Penn-State Worry Questionnaire, Ruminative Responses Scale, and Spielberger State-Trait Anxiety Inventory – Trait Subscale) is justified well. However, it was unclear to me why the authors did not include pain-related measures of catastrophizing to compare against their general catastrophizing scale, given that pain-related version were the only type to exist prior to the authors' new measure. Additionally, it was unclear why the authors chose schizotypy and positive affect scales to compare for assessing discriminant validity of their measure—why not measures of other cognitive distortions such as dichotomous (i.e. black-and-white) thinking? I think the authors could provide a more detailed rationale for why they chose the measures they did.

3) My third concern is similar to (2), but on the rationale for choosing the statistical models that the authors used to conduct their analyses. Across studies 3-4, the authors use many different statistical models to test various aspects of their research question (i.e. heterotrait-monotrait ratio of correlations for assessing discriminant validity, EFA and CFA to assess convergent validity), and I think they could explain their choice of model a bit more. For example, the authors note: *"To assess discriminant validity, we also used the heterotrait-monotrait ratio of correlations (HTMT) (21), for which statistic a value of <.85 can be considered discriminant"* (p. 12), but they do not describe why they chose this method. Additionally, I was confused at times as to why some methods were correcting for attenuation and others were not (e.g., Figure 1b). I think further clarification on these types of decisions would be useful to contextualize the authors' choices.

Minor concerns:

- 1) The authors note that their measure is able to predict diagnostic status throughout the text, and at times it is easy to miss that it is self-reported diagnostic status and not an actual clinical-interview style diagnosis. I think this could be made clearer by describing it as “self-reported diagnostic status” throughout.
- 2) The authors report that a “random subset” of participants completed the test-retest portion of the study, but also note that these participants were “first-come-first-serve”. I think they authors should clarify that these participants are not actually a random sample, in the statistical sense, to avoid confusion.
- 3) On page 24 in the discussion, the authors interpret their finding of the catastrophizing scale items loading on a single factor to indicate that *“catastrophizing is not just an epiphenomenon or straightforward consequence of anxiety and depression, but may be a separable construct with at least partially independent aetiology”*. I think this is too strong of a conclusion, or at least it assumes that there is a single underlying “catastrophic thinking” factor that causally drives people’s behavior. I do not think that it is problematic that the authors state their opinion on this matter, but instead think it is worth noting that such a conclusions makes some rather strong assumptions (in my opinion) regarding: (1) the nature of the psychological construct, and (2) the factor analytic model being an accurate representation of how the construct produces observed data. Armstrong (1967; <https://doi.org/10.1080/00031305.1967.10479849>) provides a good and concise example of why I think this interpretation can be problematic.

Appendix B

Dear Dr Emma Hayiou-Thomas

We thank you and the reviewers for your very helpful comments and suggestions. Below we present the comments and provide our responses in green italics. The edited sections of the manuscript are highlighted in yellow. We very much hope that you find the manuscript now suitable for publication in Royal Society Open Science.

Best wishes,

Alex Pike, Jade Serfaty and Oliver Robinson

Associate editor: Dr Emma Hayiou-Thomas

I am in full agreement with both reviewers that this is a strong paper, filling an important gap in the literature, using a systematic set of rigorous and well thought-through studies. The open-science approach you've taken was also highlighted as a notable strength.

We are very grateful to the associate editor for these kind comments.

The reviewers have provided a set of thoughtful and constructive suggestions for further improvement of the manuscript. I'd particularly like to highlight two points:

- 1) the request for clarification of how catastrophising is conceptualised in your scale: which level of variable you see it as (symptom, cognitive process, or diagnosis; R1)

We see catastrophising primarily as a cognitive process, and have thus removed mentions to catastrophising as a symptom in line with the reviewers' comments (see below). However, we believe that in some cases it can also be a symptom of anxiety and depression – not in the sense that it is either necessary or sufficient for a diagnosis, or indeed specific to a particular psychiatric diagnostic category, but in the sense that it is a mental phenomenon which is both present in the 'healthy' population, and may cause distress and be a risk/maintaining factor for more clinically-relevant disorder states. We do not conceive of it as a diagnosis. We have added these ideas to the introduction (lines 41-47):

'Catastrophising, for the purposes of this paper, is considered to be a cognitive process, which is more common in clinical populations (e.g. those suffering from anxiety or depression), but also exists in the general population. Catastrophising is also likely to cause distress to the individual experiencing it, and – if this is in the context of a psychiatric diagnosis – may be considered a symptom. We do not consider it to be a diagnosis, but rather a transdiagnostic factor which acts as a predisposing or maintaining factor for anxiety, depression, and perhaps other disorders.'

- 2) and the question of unidimensionality, (R2)

We agree this is an important issue, and have responded to R2 as follows:

We agree with the reviewer that finding two factors, one of which related to both the 1) overestimation of the probability of catastrophic events and 2) overestimating how bad a negative event would be (perceiving it as subjectively catastrophic). We initially had no strong hypotheses about the structure of catastrophising, as we equally thought it plausible that different 'domains' or 'contexts' might play a role – e.g. catastrophising about health, catastrophising about money, or catastrophising about the social world (similarly to the item on catastrophising included in the CDS).

To avoid overfitting, we do not believe it is sensible to perform a CFA on the same data we have already performed an EFA on (see Fokkema & Greiff, 2019, <https://doi.org/10.1027/1015-5759/a000460>), but do believe this is an important question. As such, we have added the following to the discussion (lines 545-552):

'This finding is a little surprising given the definition of catastrophising that we propose above, which includes two components – the 1) overestimation of the probability of catastrophic events and 2) overestimating how bad a negative event would be (perceiving it as subjectively catastrophic). We did not specifically test this hypothesis using a CFA in our data, and it would not be wise to do so after performing an EFA (30), but this may be a fruitful avenue for future work. It is possible that a wider item pool, with more items specifically referring to each separate component, would have shown such a two-item factor.'

- 3) a more explicit comparison of your scale with the small sets of comparable items used in other scales in the literature (e.g. in the context of pain catastrophising).

This is a useful point, which we have now addressed. We have added the following to the discussion, to explicitly compare our item bank to those used in pain catastrophizing (lines 593-599):

'The items on the Catastrophising Questionnaire are related to items used in pain catastrophising measures – indeed, we used pain catastrophising scales as inspiration for our bank of items. However, many of the items included in pain catastrophising scales either assume a current experience of pain - for example 'there's nothing I can do to reduce the intensity of the pain' – or at least assume a current difficult experience – for example 'I worry all the time about whether it will end'. Our questionnaire should be applicable to the general population, as well as psychiatric populations, by not presupposing these points.'

In addition, we also included the following in the introduction, in response to reviewer 1's query about why an additional non-pain measure is needed (lines 102-116):

'In particular, shorter questionnaires will not allow a sufficient range of scores to enable improvement or deterioration to be tracked over time (as might be required if one wished to assess the effects of decatastrophising), and also prohibit more nuanced analyses of the structure of catastrophising (i.e. the underlying factor structure). More specifically, existing self-report questionnaires which contain catastrophising items include the Cognitive Emotion Regulation Questionnaire, or CERQ (19), which includes four items to assess catastrophising, all of which are focused on the magnification of past events, in contrast to our conceptualisation of catastrophising as being future-oriented. Another questionnaire, the Cognitive Distortions Scale(20), contains two items to measure catastrophising (one in a social, and one in an achievement setting), but relies on participants' abilities to generalise from a vignette to other situations in everyday life. Other questionnaires containing items that measure catastrophising include the Children's Negative Cognitive Error Questionnaire (21) and the Children's Cognitive Style Questionnaire(22), which are not validated for use in adults.'

Reviewer: 1

Comments to the Author(s)

Overall comment:

The authors present a comprehensive set of analyses assessing the psychometric properties of a newly

developed self-report measure of catastrophising. This is an impressive and detailed body of work in an area surprisingly lacking in extant measures. In particular, the authors are to be commended for their comprehensive analyses and their open science approach to their materials and code. This is a well-written paper and overall provides a good overview of the work conducted. However, there are a few areas requiring improvement – importantly regarding the rationale for the work presented and interpretation of findings. These are detailed below, along some more minor comments.

Many thanks to the reviewer for these comments and your detailed feedback. We address all the comments in turn below.

Major comments:

- While the introduction is largely clear and well-written, there appears to be a gap in the rationale for the current work. The authors state there are few existing measures and those that do exist focus on pain-related catastrophising. However, they go on to state that existing self-report questionnaires do have some limited items on catastrophising (p.5). More detail here would be useful. What are these scales? What are the limitations of these measures? Why is a full scale needed? (& how does the limited number of items used in previous work sit with the development of a 4-item short version in the current work?)

We have now added several extra sentences to the introduction that hopefully will clarify why an additional self-report measure is necessary (lines 102-116).

‘In particular, shorter questionnaires will not allow a sufficient range of scores to enable improvement or deterioration to be tracked over time (as might be required if one wished to assess the effects of decatastrophising), and also prohibit more nuanced analyses of the structure of catastrophising (i.e. the underlying factor structure). More specifically, existing self-report questionnaires which contain catastrophising items include the Cognitive Emotion Regulation Questionnaire, or CERQ (19), which includes four items to assess catastrophising, all of which are focused on the magnification of past events, in contrast to our conceptualisation of catastrophising as being future-oriented. Another questionnaire, the Cognitive Distortions Scale(20), contains two items to measure catastrophising (one in a social, and one in an achievement setting), but relies on participants’ abilities to generalise from a vignette to other situations in everyday life. Other questionnaires containing items that measure catastrophising include the Children’s Negative Cognitive Error Questionnaire (21) and the Children’s Cognitive Style Questionnaire(22), which are not validated for use in adults.’

We chose to develop a short version of our longer questionnaire not because we believe that this necessarily captures the full complexity and allows for adequate tracking of catastrophising over time, but to enable those who wish to use a shorter measure to have one that is comparable to the full version we have created here. We have added our rationale for developing a short version to the methods section (lines 226-228):

‘We developed a short version of the Catastrophising Questionnaire, as there are a number of circumstances in which researchers or clinicians might need a briefer scale which corresponds to the full version.’

- Relatedly, the statement “Studies have looked at the item content of scales measuring catastrophising, depression and anxiety and found that they were markedly similar, although intended to measure distinct constructs (9).” requires further explanation. What is the reasoning to suggest that catastrophising is actually a separable construct if previous studies have demonstrated such similarities? Are you suggesting there’s a limitation in the operationalisation of catastrophising as a

construct in this work? (or are these pain-related catastrophising measures?) Explaining these limitations more fully is important in justifying the rationale for the work presented here.

We have inserted a sentence into the text to further clarify this, but in short, the answer is both. The study (9) that we refer to suggests that there is measurement/conceptual confounding in scales designed to measure pain catastrophizing, in that the content substantially overlaps with content related to depression, although the authors conclude that there is evidence to suggest the two constructs (depression and catastrophizing) are not redundant, and indeed that catastrophizing is a predictor for future depression, even when controlling for current depression. The relevant text now reads as follows (lines 123-127):

‘Sullivan et al. examined scales measuring pain-related catastrophizing, depression and anxiety and found that they were markedly similar in content, although intended to measure distinct constructs (9). Notably, however, the authors concluded that catastrophizing is an independent construct, but that operational or measurement confounds may have resulted in this apparent redundancy (9).’

- The inclusion of an item in the short version that is not included in the long version seems unusual and is not well justified or explained. Further detail on this decision would be useful.

The item that we included was originally part of the initial item bank, but was removed in preference for items that were more specific. However, we later considered that in a short form, having more general items would be more useful than having specific items that individuals might not be able to relate to, given the limited data collected using a short form of any questionnaire. We recognise that this is an unusual choice, and so presented an alternative analysis where this item is left out (lines 419-420):

‘We include one item in the short version that did not overlap with the final version of the Catastrophising Questionnaire. When this is omitted, the correlation between full and short version scores decreases ($r_{498} = .77, p < .001$).’

We have also amended the relevant section in the discussion (where we already discuss this issue) by detailing our justification (lines 645-655):

‘The item that we included was originally part of the initial item bank, but was removed from later versions of the questionnaire as other items were more specific. However, when developing a short form, we theorised that more general items would be more valuable, given the limited data collected using a short form of any questionnaire.’

- I am unsure of the conclusion that “catastrophising may be a construct that is independent from anxiety, worry, rumination, and depression, at least in our data.” On p.18, this refers to figure 1a which shows separation from other factors labelled ‘worry’, ‘rumination’, ‘low mood and anxiety’, ‘trait anxiety’ and ‘other’. I think the description here might best be limited to these specific variables (especially given the fairly high correlations between ‘anxiety’ (GAD-7) and ‘depression’ (PHQ-9) shown in Fig 1B). Similarly, on p.23 – this might be rephrased to more accurately reflect the findings.

This is a good point and we have replaced the quoted sentence with the following (lines 440-441):

‘Therefore, these results suggest that catastrophising may be a construct that is independent from worry, rumination, and low mood and anxiety, at least in our data.’ Furthermore, on p23 (now p25), we have amended the relevant sentence to ‘Catastrophising is independent from convergent constructs such as anxiety and low mood, worry, and rumination.’

- I find the phrasing ‘a unitary construct that is separable from anxiety, depression and worry’ to be somewhat confusing. Throughout, it is unclear if the authors consider catastrophising to be a symptom of anxiety or more of a ‘process variable’, a trait-level cognitive process which might contribute to development or maintenance of anxiety or depression (similar to intolerance of uncertainty, or anxiety sensitivity for example; Norton et al., 2005). These process variables would be considered at a different level to either diagnoses (anxiety, depression) or symptoms (e.g., worry). It might be anticipated that higher levels of catastrophising tendencies may manifest as more frequent worry (symptom) and contribute to anxiety or depression (diagnosis). Alternatively, if they consider catastrophising to be a specific symptom that is not adequately measured in existing diagnostic questionnaires, this should be addressed in the manuscript. Clarity on the author’s conceptualisation of catastrophising may then help to make some of the more ambiguous phrasing (such as that specified above) more concrete.

We thank the reviewer for this comment, and are grateful for the opportunity to think more deeply about these definitions and clarify this paper further. We believe that catastrophising is a cognitive process, akin to intolerance of uncertainty or anxiety sensitivity. In particular, we would hypothesise that it is a cognitive pattern, of sorts, which may be more present in certain environmental conditions, such as where there are high costs to underestimating a situation, or where there are outcomes that occur rarely but have a large associated cost.

We would argue, however, that symptoms/manifestations of a disorder and predisposing cognitive processes are not necessarily separate, though this perhaps depends on the definition of symptoms. If the reviewer means this either a) in the sense of a maintaining vs. predisposing factor, or b) in the sense of an observable manifestation of a disorder that causes distress vs. a process that underpins distressing symptoms; in both cases we would also suggest catastrophising is both. To this extent, we would therefore also conceptualise catastrophising as a symptom. However, we do not believe that catastrophising is specific to any given disorder, but that elevated catastrophising might be found at a higher rate in those with a diagnosis of anxiety or depression (and likely other mental health conditions). To this extent it is akin, in some ways, to negative affective bias (i.e., negative schemata) – a cognitive process which is present more in those with anxiety or depression (and indeed promotes and upholds negative mood state in those with meeting clinical thresholds), but is also present in the general population. We do not believe, however, that catastrophising on its own is a diagnosis – rather a trans-diagnostic process that can contribute to other diagnoses.

However, we recognise that it is much more straightforward to refer to catastrophising as a cognitive process in this paper to avoid semantic confusion. We have therefore removed any mention of catastrophising as a symptom, and now refer to it as a cognitive process throughout. We also summarise the above argument in the introduction (lines 41-47):

‘Catastrophising, for the purposes of this paper, is considered to be a cognitive process, which is more common in clinical populations (e.g. those suffering from anxiety or depression), but also exists in the general population. Catastrophising is also likely to cause distress to the individual experiencing it, and – if this is in the context of a psychiatric diagnosis – may be considered a symptom. We do not consider it to be a diagnosis, but rather a transdiagnostic factor which acts as a predisposing or maintaining factor for anxiety, depression, and perhaps other disorders.’

- Regarding discriminant validity – only the TEPS scores are specifically stated as showing discriminant validity (p.20). Was there a definition of ‘convergent’ validity? Do the authors consider the associations with GAD/STAI/PSWQ/RRS to be convergent or discriminant here? From the discussion of these findings, the authors consider catastrophising to be separable from

anxiety/depression etc. (p.23), but also positively related to these measures (p.24). Again clarity on the types of variables involved might help improve the interpretability of these statements (e.g., catastrophising as a process variable related to anxiety and depression, but clearly loading onto a separate latent factor compared to worry/rumination etc. – or catastrophising as a separate symptom cluster??)

We considered ‘convergent’ validity to be demonstrated when there was a significant positive association between constructs that we hypothesised were related (though not redundant or identical). We have added a section discussing the definitions we use into the text (lines 241-249):

‘We collected data on a number of additional self-report questionnaires, in order to assess convergent and discriminant validity. We define construct validity here as positive associations between two measures of the same construct (i.e. two different questionnaires measuring the same thing), and define convergent validity as positive (but not large enough to indicate redundancy) associations between two measures of different constructs which are theoretically hypothesised to be related. We define discriminant validity as a lack of significant positive association between measures of constructs that are not hypothesised to be related to catastrophising, but are related to mental health – such as anhedonia, schizotypy and alcohol use disorder. These measures are itemised below.’

Based on these definitions, we consider associations with the GAD/STAI/PSWQ/RRS to reflect convergent validity, and to both be separable (i.e. not redundant) and yet have a significant positive relationship, much as is seen for example with height and weight. Hopefully this, in line with the response to the previous comment, clarifies our stance on this issue a little further and allows greater interpretability – in short, we hypothesise that catastrophising is a process variable related to anxiety and depression, but loading onto a separate latent factor.

We have also moved text that was previously in the supplementary materials, justifying the use of the convergent/discriminant measures we chose, into this section, in line with your subsequent comments and to add clarity to this section (lines 250-290):

‘Construct validity measures

Cognitive Distortion Scale: Catastrophising Subscale

The Cognitive Distortion Scale (CDS) conceived by Covin et al. contains a 2-item Catastrophising subscale (Covin et al., 2011). This subscale was used for comparison with our Catastrophising Questionnaire to assess construct validity.

Convergent measures

Anxiety

Catastrophising is believed to relate to psychiatric diagnoses such as anxiety and depression, and as such should be related to scores on questionnaires measuring symptoms of these disorders. Therefore, we included the GAD-7 (Generalized Anxiety Disorder Assessment), which measures symptoms of generalized anxiety disorder over the previous two weeks (Spitzer et al., 2006). We also included the STAI-T (Trait Anxiety Subscale of the State Trait Anxiety Inventory) which measures anxiety level as an enduring personality trait, and focuses on anxiety in general rather than on a specific anxiety disorder (Spielberger et al., 1983).

Depression

For the same reasons as above, we included the PHQ-9 (Patient Health Questionnaire) which is a depression scale that asks about symptoms over the preceding two weeks (Spitzer et al.,

1999). As we were testing participants online and could not assess risk of harm in person, we removed the question about suicidal ideation.

Worry

Worry is an important feature of anxiety disorders that may also be related to catastrophizing. Thus, we included the PSWQ (Penn State Worry Questionnaire) which measures traits of worry independently from anxiety (Meyer et al., 1990). It examines the excessiveness, generality and uncontrollable dimensions of worry.

Rumination

Rumination is related to depression, and we thus hypothesised it would also correlate with catastrophizing. However, it is thought to be separable from depression, so in addition to the PHQ-9, we included the RRS (Rumination Response Scale) which measures two aspects: brooding and reflective pondering (Nolen-Hoeksma and Morrow, 1991).

Discriminant measures

Experience of Pleasure

We expected the experience of pleasure to be orthogonal to catastrophizing as it implies a positive conception of the anticipation of pleasure and enjoying the present moment. Therefore, we included the TEPS-ANT and TEPS-CON (Temporal Experience of Pleasure Scale – Anticipatory and Consummatory subscales), which assess pleasure experienced during anticipation of reward and on attaining rewards, respectively (Gard et al., 2006).'

- Regarding the incremental validity findings. Do the authors have an interpretation for these findings? Why might a measure of catastrophizing improve prediction of diagnosis and medication usage? Is this simply a sensitivity finding (i.e., with an additional measure you can capture more variance?) – or do the authors think there are aspects of catastrophizing that are particularly strong indicators of psychiatric diagnoses and medication usage?

This is a good question. We believe that catastrophizing is a cognitive process that acts as both a risk and maintaining factor for diagnoses such as anxiety and depression. We add the following to the text of the discussion, where this is mentioned, and have removed the section from the paragraph above that previously highlighted the clinical uses of this scale (lines 586-591):

'We believe, therefore, that our measure of catastrophizing captures an important additional risk factor for diagnosis that is not present in existing scales. From a clinical perspective, we believe that having an ability to measure general catastrophizing will facilitate the assessment of the efficacy and mechanism-of-action of de-catastrophizing approaches, and, furthermore, items on the scale may potentially provide patient-specific directions for de-catastrophizing during therapy.'

Minor comments:

Abstract:

- Might be helpful to describe catastrophizing as a cognitive process in the first sentence (as in para 2 of the introduction)

We have inserted this into line 1 of the abstract:

‘Catastrophising is a cognitive process that can be defined as predicting the worst possible outcome.’

- The final sentence states catastrophising is a psychiatric symptom – as mentioned above, clarity on symptom vs. process level variable would be useful

Apologies for our lack of clarity here. We have reframed this to read (lines 20-21):

‘Critically, we can now, for the first time, obtain detailed self-report data on catastrophising’.

Methods:

- Power calculations are discussed in the limitations section, but not mentioned in the methods and don't appear to be in the supplementary materials?

We highlight in the limitations section that we used rules of thumb rather than a power analysis. The aim of the discussion of power analysis in this context was designed to justify this choice by detailing the limitations of trying to perform a power calculation in factor analytic research.

However, we recognise that this is unclear and detracts from the main message in this paragraph, and have therefore moved the following line to the methods and edited it slightly (lines 173-177):

‘We did not perform a power analysis to determine the sample size for these studies, as the issue of how best to perform a power analysis for factor analysis research is still an open question: it depends on the nature of the data being collected and the model it is fitted to, and even the best methods require the researcher to make a number of assumptions about the population values of estimated parameters and/or have initial data (24,25).’

- Quality assurance checks are mentioned for study 3, but not for studies 1 or 2 – were these included only for study 3? Is there a reason for this?

We thank the reviewer for the opportunity to clarify this point. We have added the explanation for this choice to the supplementary materials (lines S47-S50):

‘We did not perform these checks for earlier studies as these were preliminary studies to allow us to refine the questionnaire items. Additionally, it was only in study 3 that we included repeated items (in the short form and full form of the questionnaire), so we could only analyse divergence between these responses here.’

- The ordering of justification of the additional scales could be improved. After reading the entire paper it is clear what the rationale for including each measure is, but when these first appear in the methods, this is unclear. It might be prudent to review the ordering of this information – and at least describe in a brief, broad overview what the purpose of these scales is before they are mentioned in detail.

We have moved the text discussing the justifications for including the additional scales that was previously in the supplementary materials into the main text (lines 250-290), which hopefully will ensure that the reader understands this on the first read-through and does not have to go hunting for the justification.

- Results of the Lavaan CFA in the supplementary materials are described in the methods section – these would be better placed in context in the results section

Many thanks for this comment. We now add the following to the results section (lines 450-454):

When the same analysis was performed in a structural equation modelling framework, in order to disattenuate for measurement error, all the correlations increased, but this was not specific to correlations with the Catastrophising Questionnaire alone (Supplementary Figure 7). This provides further evidence for convergent validity without redundancy.

- Findings reported in the results section could use more context (e.g., p.15 – ‘All indices are presented in Table 1’; it is not immediately apparent what this refers to)

We have replaced this phrase with the following each time it appears (lines 366, 387, 498, 513):

‘Cronbach’s alpha, omega, and the inter-item and inter-total correlations’.

- Table 1 – it would be useful to provide for the reader the definition of ‘reliable’ and ‘valid’ based on the values reported;

The following text has been added to the caption for Table 1 (lines 370-376):

‘Internal consistency is measured by Cronbach’s alpha, Omega, and inter-item correlations. Item-total correlations are used to assess discriminability of items – i.e. the extent to which they distinguish between high and low scorers on a task. Whilst adequate values for these measures depends on the application, the nature of the construct, and other factors, some suggested values for acceptability are given to aid interpretation. 0.8 (for basic or applied research) is considered to be an acceptable value for Cronbach’s alpha, with .9 or above recommended for applied work that is guiding decision-making (25,26). Similar values are recommended for Omega. Mean inter-item correlations should be between .15 and .5 (25). Item-total correlations should be over 0.3 or 0.4 (27).’

- adding a brief summary of the scale used (e.g., catastrophising scale – 24 item version) to the column currently listing ‘study 1’ etc. would be useful

We have added a column to this table entitled ‘Questionnaire version’ (line 376 on), and list the item number as an identifier.

- The supplementary materials reports that “in all our studies, the participants who gave us feedback generally judged the Catastrophising Questionnaire to be clear, relevant to their experiences, and easy to complete” – if this is to be reported, more detail on the types of questions and mean scores should be included to allow the reader to assess this statement for themselves

After participants had completed the latest version of the questionnaire, we asked them ‘Please let us know if any of the above statements are unclear and if you have any comments you would like to add’ and presented them with a free text box to write their thoughts. We did not perform any systematic analyses of these free text responses, so we have removed this statement from the supplementary materials.

- The statement ‘whether clinicians thought an item captured elements of catastrophising’ requires further explanation (which clinicians – what expertise? How many? Was this a formal or informal process?)

We informally met with mental health clinicians (three qualified clinical psychologists) and asked for their opinions on the questionnaire items. We have now attempted to clarify this within the text (line S94):

‘mental health clinicians (who we informally consulted during this process)’

Additional reference:

Norton, P. J., Sexton, K. A., Walker, J. R., & Ron Norton, G. (2005). Hierarchical model of vulnerabilities for anxiety: Replication and extension with a clinical sample. *Cognitive Behaviour Therapy*, 34(1), 50-63.

Reviewer 2:

Summary:

In the current study, the authors developed a self-report scale to measure catastrophizing, which is comprised of two facets including: predicting an objective catastrophe will occur to oneself (e.g., a heart attack, which others agree would be catastrophic), or having the subjective perception that a negative/unwanted outcome will be catastrophic (e.g., social rejection being perceived as catastrophic, when most other people would not assume so). In both cases, catastrophizing involves assuming that the worse will happen.

The authors chose to focus on catastrophizing due to its prominence in cognitive behavioral theories of anxiety, depression, and other forms of psychopathology (i.e. it is one of many “cognitive distortions” that is often present transdiagnostically). As they mention, “decatastrophizing” is a key component of cognitive behavioral therapy, as catastrophizing is thought to play a causal role in the persistence of anxiety and avoidance. Importantly, there are no self-reports of catastrophizing outside of the context of pain catastrophizing, which is an important oversight given its clinical relevance.

Given the lack of self-reports available for measuring catastrophizing, the authors developed and validated their own pool of items to do so. They use multiple samples to refine their item pool, test for convergent and discriminant validity, and develop a short-form version of the full scale. The authors show that a 24-item full version and a 4-item short version scale both exhibit acceptable fit statistics (according to a single factor, factor analytic model).

Strengths:

I think that the authors chose a really important topic, and they identified a clear gap in the literature that I was surprised to learn that no previously existing measure was developed to fill. Catastrophizing is indeed a key component of cognitive behavioral theories and treatments of psychopathology, and the availability of a measure should be very useful for researchers and clinicians alike (particularly the short version in the case of clinical application). Additionally, the authors conduct a rather comprehensive set of analyses to show how their catastrophizing measure converges with other similar constructs and diverges from those that we would not expect it to correlate with. Finally, the authors make their code and self-report items available online, which is very beneficial to the research community.

We are very grateful to this reviewer for their positive comments and their accurate summary of our paper. We address their comments below, in turn.

Major concerns:

Overall, I think this was a really good study and it provides an important addition to the literature. I have a few concerns that I think should be addressed before publication of this

manuscript, although none require substantial changes to the analyses or text. My most important concerns are as follows:

1) My first concern is regarding the unidimensionality of catastrophizing. For example, as described by the authors in the introduction and noted above in my summary, two facets including: (a) predicting an objective catastrophe will occur to oneself; and (2) having the subjective perception that a negative/unwanted outcome will be catastrophic, fall under the general definition of catastrophizing. In clinical settings, these two facets are often discussed to reflect: (a) overestimation of the probability of negative events occurring; and (2) underestimation of ability to cope with a negative event that occurs, or overestimating how bad the negative event would actually be (e.g., Foa & Kozak, 1986; 10.1037/0033-2909.99.1.20; McNally & Foa, 1987; 10.1007/BF01183859). Although these are related, they are distinct in that one involves estimating event probabilities, and the other costs/values. I think some discussion of these ideas could be important to include in the manuscript. For example, would the authors expect to find multiple factors if they had a wider item pool to assess these different mechanisms to start with? Looking at the items that the authors included, it looks like they do cover both probability and value estimation/valuation. They could potentially test this idea using CFA, but I think it would at least be worth discussing in the context of their 1 factor model.

We agree with the reviewer that finding two factors, one of which related to both the 1) overestimation of the probability of an event and 2) overestimating how bad the negative event would be. We initially had no strong hypotheses about the structure of catastrophizing, as we equally thought it plausible that different ‘domains’ or ‘contexts’ might play a role – e.g. catastrophizing about health, catastrophizing about money, or catastrophizing about the social world (similarly to the item on catastrophizing included in the CDS). We do not believe that performing a hypothesis-driven CFA on the same data we have already performed an EFA on (to test convergent validity) is justified (Fokkema & Greiff, 2019, <https://doi.org/10.1027/1015-5759/a000460>), but do believe this is an important question. We have added the following to the discussion (lines 545-552):

‘This finding is a little surprising given the definition of catastrophizing that we propose above, which includes two components – the overestimation of the probability of rare negative events, and the overestimation of the magnitude or severity of negative events. We did not specifically test this hypothesis using a CFA in our data, but this may be a fruitful avenue for future work. It is possible that a wider item pool, with more items specifically referring to each separate component, would have shown such a two-item factor.’

2) My second concern is partly due to the formatting/organization of the text, but also a conceptual concern. I had trouble following the rationale for why the specific scales in Table 3 (which appears before Tables 1 and 2 in the text, which partly explains my confusion) were included to test the convergent and discriminant validity of the new Catastrophizing Scale. The authors note that they include the PHQ-9 and GAD-7 scales due to their widespread use for measuring general anxiety and depression in clinical settings in the UK (which I think is a solid rationale). Also, their inclusion of rumination, trait anxiety, and worry related measures (i.e. the Penn-State Worry Questionnaire, Ruminative Responses Scale, and Spielberger State-Trait Anxiety Inventory – Trait Subscale) is justified well. However, it was unclear to me why the authors did not include pain-related measures of catastrophizing to compare against their general catastrophizing scale, given that pain-related version were the only type to exist prior to the authors’ new measure. Additionally, it was unclear why the authors chose schizotypy and positive affect scales to compare for assessing discriminant validity of their measure—why not measures of other cognitive distortions such as dichotomous (i.e. black-and-white) thinking? I think the authors could provide a more detailed rationale for why they chose the measures they did.

Thank you for pointing out our oversight – Table 3 was indeed supposed to be in the results section, within the ‘Study 3’ subsection. We have moved this table accordingly.

We did not include pain-related measures of catastrophizing as many of them would have seemed inappropriate and confusing to a general audience of participants completing them online, such as ‘I pray for the pain to stop.’ (from the CSQ) or ‘There’s nothing I can do to reduce the intensity of the pain’ (from the PCS). We could perhaps have asked participants to imagine a situation in which they were in pain, but this would most likely have been confusing to participants and their responses would have probably depended on other factors, such as their experience (or lack) of chronic pain, or their ability to conceptualise a painful experience. We have added a mention of this to the limitations section of our discussion (lines 608-611):

‘Relatedly, we did not include pain catastrophizing measures in our study of convergent validity, which may have given us valuable additional information. Our justification for this was that many of the questions in pain catastrophizing scales explicitly discuss a current pain experience, making them inappropriate to administer to a general sample of the population.’

We did not include measures of other cognitive distortions as examples of discriminant validity as we would hypothesise that, rather than having no relationship with catastrophizing, all unhelpful coping strategies are related to a certain extent. For example, an individual who catastrophises might also show a tendency to jump to conclusions, or might also generalise. However, we do believe that it will be important to study how these different cognitive distortions are related in future work, and have added a line to the discussion emphasising this (lines 565-568):

‘This work could be further extended, by also testing whether catastrophizing is independent from other cognitive distortions such as black-and-white thinking or overgeneralisation.’

We chose to include measures that we thought were theoretically completely independent of catastrophizing, but still related to mental health such as the TEPS, a schizotypy scale, and an alcohol use disorder scale. We now state this in the methods section (line 246):

‘We define discriminant validity as a lack of significant positive association between measures of constructs that are not hypothesised to be related to catastrophizing, but are related to mental health – such as anhedonia, schizotypy and alcohol use disorder.’

3) My third concern is similar to (2), but on the rationale for choosing the statistical models that the authors used to conduct their analyses. Across studies 3-4, the authors use many different statistical models to test various aspects of their research question (i.e. heterotrait-monotrait ratio of correlations for assessing discriminant validity, EFA and CFA to assess convergent validity), and I think they could explain their choice of model a bit more. For example, the authors note: “*To assess discriminant validity, we also used the heterotrait-monotrait ratio of correlations (HTMT) (21), for which statistic a value of $<.85$ can be considered discriminant*”(p.12), but they do not describe why they chose this method. Additionally, I was confused at times as to why some methods were correcting for attenuation and others were not (e.g., Figure 1b). I think further clarification on these types of decisions would be useful to contextualize the authors’ choices.

We chose to examine convergent validity by use of an EFA as a model-free way of attempting to characterise the relationships within the items – rather than assuming that each questionnaire is separate, we wanted to test whether the catastrophizing questionnaire items coupled together with other questionnaires, or whether they seemed to load separately onto a factor of their own (as was the case). We chose to use the HTMT as we believe this to be a particularly robust technique for the analysis of divergent validity (e.g. <https://doi.org/10.1108/IntR-12-2017-0515>, <https://doi.org/10.1007/s11747-014-0403-8>). We corrected for attenuation in an exploratory analysis, presented in the supplementary materials, in order to assess whether the anticipated heightened

correlations between items was specifically increased for correlations with the catastrophising questionnaire, which might have indicated redundancy with other questionnaires. We have added these explanations into the text (lines 297-304, and 309-314):

‘We chose to use an EFA as a model-free way of attempting to characterise the relationships between individual items from all questionnaires – rather than assuming that each questionnaire is separate, we wanted to test whether the catastrophising questionnaire items coupled together with other questionnaires, or whether they loaded separately onto a factor of their own. To assess discriminant validity, we also used the heterotrait-monotrait ratio of correlations (HTMT), which is a particularly robust measure of discriminant validity (24), for which statistic a value of $<.85$ can be considered discriminant.’

‘We anticipated that this would result in heightened correlations between questionnaires, due to the disattenuation for measurement error for each indicator. The aim of this analysis was to assess whether the anticipated heightened correlations between items were specifically increased for correlations with the catastrophising questionnaire, which might have indicated redundancy with the other questionnaire measures we used.’

Minor concerns:

1) The authors note that their measure is able to predict diagnostic status throughout the text, and at times it is easy to miss that it is self-reported diagnostic status and not an actual clinical interview style diagnosis. I think this could be made clearer by describing it as “self-reported diagnostic status” throughout.

Many thanks for this comment – we agree that it was not clear as it was. We have now amended this to state:

‘self-reported diagnostic status’ throughout (the ‘Study 3’ sections under methods and results, the discussion, and the caption for supplementary table 1).

2) The authors report that a “random subset” of participants completed the test-retest portion of the study, but also note that these participants were “first-come-first-serve”. I think they authors should clarify that these participants are not actually a random sample, in the statistical sense, to avoid confusion.

The reviewer is indeed correct that this sample was not truly random – we have rephrased to (line 336):

‘an unselected, first-come, first-served group’.

3) On page 24 in the discussion, the authors interpret their finding of the catastrophizing scale items loading on a single factor to indicate that “*catastrophizing is not just an epiphenomenon or straightforward consequence of anxiety and depression, but may be a separable construct with at least partially independent aetiology*”. I think this is too strong of a conclusion, or at least it assumes that there is a single underlying “catastrophic thinking” factor that causally drives people’s behavior. I do not think that it is problematic that the authors state their opinion on this matter, but instead think it is worth noting that such a conclusions makes some rather strong assumptions (in my opinion) regarding: (1) the nature of the psychological construct, and (2) the factor analytic model being an accurate representation of how the construct produces observed data. Armstrong (1967; <https://doi.org/10.1080/00031305.1967.10479849>) provides a good and concise example of why I think this interpretation can be problematic.

We agree that perhaps our opinion here is stated a little strongly. We have added a caveat to this sentence, which hopefully alleviates this concern as well as citing the paper mentioned by the reviewer (574-575):

‘Therefore, we suggest that catastrophising is not just an epiphenomenon or a straightforward consequence of anxiety and depression, but may be a separable construct with at least partially independent aetiology (although, notably, factor analytic methods may not always accurately capture the underlying factor structure of a construct (31)).’